# Operando investigation of the synergistic effect of electric field treatment and copper for bacteria inactivation

Mourin Jarin[1], Ting Wang[1] & Xing Xie [1,2] ✉

As the overuse of chemicals in our disinfection processes becomes an ever-growing concern, alternative approaches to reduce and replace the usage of chemicals is warranted. Electric field treatment has shown promising potential to have synergistic effects with standard chemical-based methods as they both target the cell membrane specifically. In this study, we use a lab-on-a-chip device to understand, observe, and quantify the synergistic effect between electric field treatment and copper inactivation. Observations in situ, and at a single cell level, ensure us that the combined approach has an enhancement effect leading more bacteria to be weakened by electric field treatment and susceptible to inactivation by copper ion permeation. The synergistic effects of electric field treatment and copper can be visually concluded here, enabling the further study of this technology to optimally develop, mature, and scale for its various applications in the future.

Microbial inactivation remains one the most important processes in our modern-day societies as the development of disinfection methods for food preservation, surfaces cleaners, and drinking water treatment has continued to limit the spread of infection and disease in our human population. Most of these processes rely on chemicals such as chlorine and antibiotics, but the heavy usage of these over the last century has now exposed some drawbacks. For the past 100 years, chlorine-based disinfection has been the standard method for reliable and cost-effective treatment of our drinking water, but over the last 50 years, it has been discovered that chlorine contributes to the formation of disinfection by products (DBPs), which have been evidenced to be carcinogenic[1,2]. Antibiotics have also been undoubtedly successful in the fight against bacterial infections, saving millions of lives over the years[3]. Nevertheless, the number of infections caused by multi-drug resistant (MSR) bacteria is increasing and antimicrobial resistance (AMR) caused by various chemical substances is presenting to be a significant threat and challenge to the environment and healthcare systems worldwide[3-7]. The World Health Organization (WHO) reported in 2019 that AMR has been responsible for the death of 700,000 people, while this number may increase to 20 million by 2050[8]. Because of the challenges and concerns associated with current chemical disinfection approaches, it is necessary to continue developing more widely applicable, safe, and sustainable alternatives for microbial inactivation. This has increased the motivation to implement the use of more physical processes to replace and reduce the usage of chemical disinfectants and antibiotics, minimizing the formation of DBPs and AMR or eliminating the effects altogether.

Electric field treatment (EFT) is an emerging technology for food processing and drinking water disinfection[9]. It is an electrophysical process capable of inactivating pathogens through electroporation, different from typical electrochemical processes that kill pathogens by direct oxidation and/or releasing microbicidal chemicals[10]. Current challenges of EFT, mainly including high voltage requirement and high energy consumption, have limited its widespread use in larger scale applications[11-15]. To address these challenges, researchers have been developing a novel process named locally enhanced electric field treatment (LEEFT), where the electrodes applied are typically modified with nanowires[16-23]. LEEFT has demonstrated great performance with low applied voltage and energy consumption (achieving ~6 log inactivation of *E. coli* using only a minimum of ~1 V and ~1 J of energy per liter of water treated)[17]. Despite the potential shown, the robustness of the nanowires on the LEEFT electrodes are not yet developed enough for

[1]School of Civil and Environmental Engineering, Georgia Institute of Technology, Atlanta, GA 30332, USA. [2]Institute for Electronics and Nanotechnology, Georgia Institute of Technology, Atlanta, GA 30332, USA. ✉e-mail: xing.xie@ce.gatech.edu

long-term use. Therefore, EFT can be a promising technology added to the toolbox of physical water disinfection techniques, but the implementation of it alone for large scale water disinfection is still not practical.

During EFT, exposing bacteria to an external electric field creates an electric potential across the cell membrane that can alter its structural integrity[10,21]. If the electric field applied is strong enough, it can cause irreversible electroporation and cell inactivation[24]. It has also been reported that with lower electric fields and smaller membrane potential changes, ion transport channels can be initiated, increasing the permeability of the cell membrane[25]. Considering that chemical disinfectants typically also attack the cell membrane to trigger the process of inactivation, researchers have applied EFT together with chemical disinfectants, aiming to achieve synergistic microbial inactivation. For example, EFT and chlorine have been used in studies to successfully inactivate bacteria through both physical and chemical methods while dramatically reducing the chlorine concentration necessary[19,23]. Huo et al. uses EFT and electrochlorination to result >6 log inactivation of bacteria and viruses with a low concentration of 0.15 mg/L chlorine[19]. This compared to conventional high dose processes using 6-15 mg/L, can drastically reduce the DBP formation[26]. In another study, EFT is combined with ozone to significantly enhance inactivation (achieving >5 log with only 0.08 mg/L $O_3$), confidently reducing the formation of bromate and other potential DBPs[27]. This potential synergism on microbial inactivation is a unique advantage of EFT compared to other physical methods, such as membrane filtration that removes whole cells and ultraviolet light (UV) that targets nucleic acids inside the cell. Majority of other physical disinfection approaches do not target only the cell membranes of pathogens, limiting the potential for two methods to work together.

Copper (Cu) is a historically well-known metal for its natural biocidal properties, inactivating microbes by damaging their cell membranes and attacking their intracellular components[28,29]. Although Cu is currently used as a natural algaecide and secondary pathogen control measure in lakes, pools and spas, plumbing systems, cooling towers, etc., it is not popularized for typical use in drinking water disinfection[28,30]. This is because high levels of Cu consumed can cause adverse health effects to humans, resulting in the US Environmental Protection Agency (EPA) setting strict guidelines for Cu in drinking water at a maximum contaminant level of 1.3 mg/L[31]. Nevertheless, Cu is not only safe at lower concentrations but also an essential nutrient for human health[32]. In our previous study, EFT was configured within a copper ionization cell, achieving ~6 log removal of *E. coli* with a very low effluent copper concentration (0.2 mg/L)[33]. This device operated continuously for 12 h, successfully treating a total of 7.2 liters of water, resulting in an estimated cost of the device at ~$10 and a cost of treated drinking water at ~$0.1/m³ [33]. This system shows promising potential for point-of-use applications in treatment systems for hospitals, emergency response after disasters, rural/remote areas, and developing/marginalized communities. Because of this, it is necessary to better understand the inactivation mechanism to further scale up and potentially implement the technology later on. Although the previous study demonstrated enhanced performance in microbial inactivation, the process for EFT and antimicrobial Cu ions working synergistically has yet to be elucidated at the cellular level.

In order to zoom in on the EFT-Cu system and observe the mechanism at work in situ, we can use a lab-on-a-chip (LOAC) device, which allows us to conduct critical mechanism studies at scales unobservable to the naked eye. In addition, we can further reduce the redundancy and tediousness of bulk electroporator based experiments that apply one condition at a time, eliminate the need for wasted materials and chemicals in larger scale studies, and result larger volumes of data with singular, microscale, well-designed experiments[34-38]. In this study, a LOAC device is fabricated and used (Fig. 1a, b) to understand, observe, and quantify the synergistic effect between EFT and Cu inactivation at microscale. The electrodes on the LOAC device have a curved edge design to generate a range of electric field strengths with a certain applied voltage (Fig. 1c, d). A wide-range of experiments on the EFT-Cu combination approach is conducted using the LOAC device with various operating parameters (e.g., pulse width, electric field strength, time, and Cu concentration) to confidently confirm the observance of a significant synergistic effect tested under these conditions. The results for overall performance of EFT-Cu, the added residual effect over time, and a single-cell analysis of the mechanism are presented and discussed. Some insight into potential implementation of this understanding to scale-up the technology, limitations of the current LOAC device, challenges with studying this technology, and future directions for optimization of EFT-Cu are also provided at the end.

## Results

### Overall Performance of EFT-Cu

The LOAC device was initially tested for its inactivation efficacy with exposure to only Cu ion solution to determine how well the device could capture the death of model bacteria from Cu ion permeation. The results showed 1 mg/L Cu can induce ~15% of cell inactivation, while 2 mg/L Cu can have ~30% (Supplementary Fig. 1), setting a baseline for inactivation with Cu using this device (refer to Supplementary Note 1 for further details). Figure 1e, f show the individual images in differential interference contrast (DIC) (Fig. 1e) and fluorescent (Fig. 1f) modes of the LOAC device after applying EFT-Cu with operating parameters of 500 ns pulse width, 500 μs period, an applied 80 V, an effective treatment time of 20 ms, and a Cu concentration dosage of 0.5 mg/L. Figure 1e demonstrates successful pre-treatment of the chip as there are no visible gaps where the bacteria are immobilized. Figure 1f demonstrates the fluorescent red, inactivated cells across the electric field gradient along the channel. The inactivation here refers to positive propidium iodide (PI) staining further explained in the methods section. The yellow rectangles drawn in Fig. 1e, f indicates a high electric field strength region where the device applies 37 kV/cm on the higher end and achieves the average inactivation percentage of ~53% (Fig. 1g). On the lower end of this highlighted region, the electric field strength applied is 25 kV/cm and achieves a much lower average inactivation percentage of ~8% (Fig. 1g). This figure examples the process in which all the following results were successfully analyzed.

### Varying Cu Concentration

Figure 2 displays the EFT-Cu combined experiments for all tested pulse widths (500 ns, 1 μs, and 2 μs from top down). The inactivation percentage over the applied electric field strength curves are displayed for increasing Cu concentration in Fig. 2a–c. Focusing on Fig. 2a, and only the most upper trend for 2 mg/L Cu (purple star curve), there is a clear scatter of values around ~35% inactivation for the first half and a rapid spike up to 100% inactivation in the second half starting around 30 kV/cm electric field strength. Looking similarly for Fig. 2b, c, the same trend is observed, where both the purple star lines average ~35% inactivation in the first half and then rapidly spike at ~27 kV/cm (Fig. 2b) and ~20 kV/cm (Fig. 2c). This indicates that under these conditions, there is a threshold of electric field strength necessary to start electroporating bacteria regardless of pulse width. After this threshold is crossed, the inactivation values increase rapidly as shown in Fig. 2a–c, where the inactivation efficiency observed is much more sensitive to changes in the electric field strength. Our detailed analysis results in consistent trends in all three conditions. For 500 ns pulse width and 0 mg/L Cu (Fig. 2a, grey hexagon curve), the lethal electroporation threshold (LET, when inactivation percentage reaches 50%) occurs at ~38 kV/cm, while for 1 mg/L Cu (blue square curve) it is ~35 kV/cm, and 2 mg/L Cu (purple star curve) achieves this at only ~31 kV/cm applied. This result

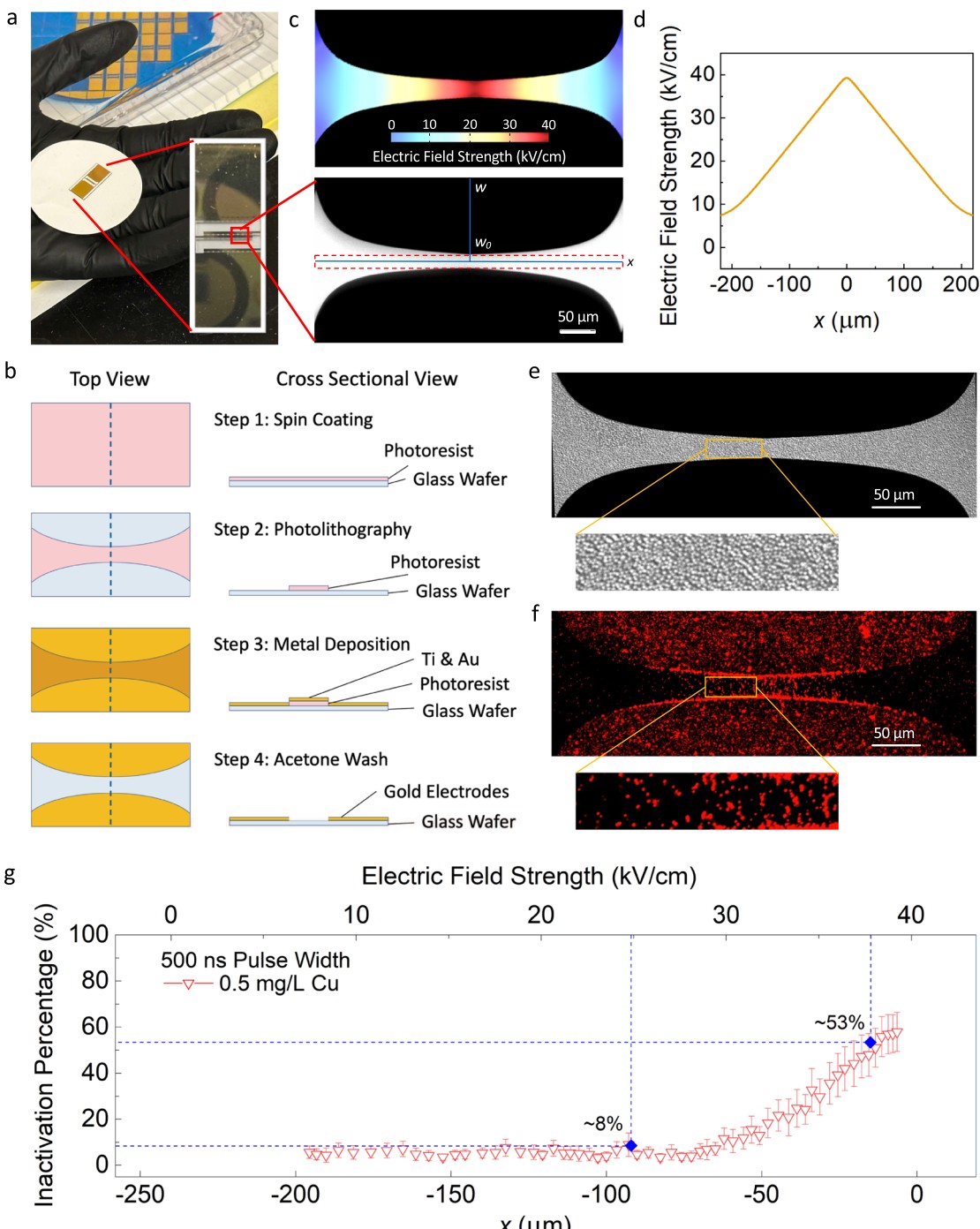

**Fig. 1 | The lab-on-a-chip (LOAC) device and its application in this study.**
Examples of images and data analysis are presented for the EFT-Cu condition with 500 ns pulse width, 500 μs period, 40k pulses, 80 V applied voltage, and a 0.5 mg/L Cu concentration. **a** A digital photo shows the LOAC device in size respect to a hand and vertical image to show the gold electrode layer. The zoomed in image shows an individual channel of the chip captured by the microscope. The curved electrode design is easily shown in respect to key parameters ($w$, $w_0$, and $x$, further explained in Supplementary Note 2). This one channel shows the scale, and the red dashed portion represents the area selected for image and data analysis consistent for all experiments. **b** The fabrication flow for both the top and cross-sectional view. The dashed line visible in the top view represents the cross-sectional view shown. **c** A COMSOL Multiphysics simulation of the electric field strength produced across an individual channel. The colored strength key is shown for reference to this example condition. **d** The plotted representation of the simulated linear electric field strength established by the curved electrode. **e** The individual differential interference contrast (DIC) channel image zoomed in with a yellow rectangle to highlight the immobilized cells that are well attached and closely packed together. **f** The individual fluorescent channel image effectively stained with the propidium iodide (PI) dye, and a zoomed in portion with a yellow rectangle to show a high electric field strength region where cells are inactivated and appear bright red. **g** The overall inactivation percentage over the x-axis and electric field strength as translated from the COMSOL Multiphysics simulation. The inactivation here refers to positive PI-staining further explained in the methods section. In all our results presented, the entire region is analyzed and averaged with the duplicate values on the second half of the x-axis. The dashed blue lines are presented to guide the reader through a typical analysis process. The error bars indicate 95% confidence intervals from the mean values based on 30 replicate channels. (For any references to colors in the figures, the reader is referred to the online/web version of this article).

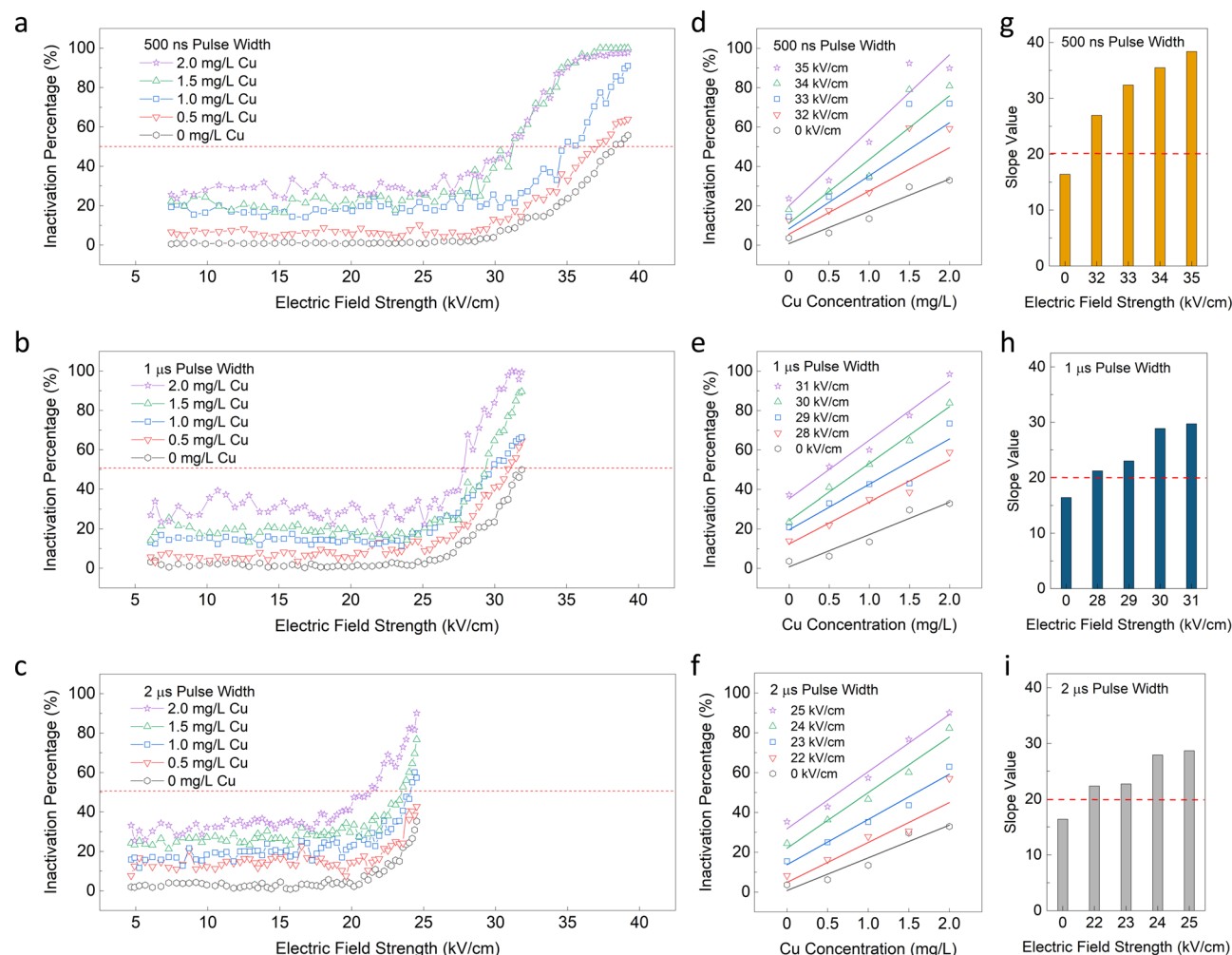

**Fig. 2 | Microbial inactivation of combined electric field treatment and copper (EFT-Cu).** The overall inactivation percentage over the electric field strength for five varied Cu concentrations 0-2 mg/L are presented when the pulse width is 500 ns (**a**), 1 μs (**b**), and 2 μs (**c**). The overall inactivation here refers to positive PI-staining further explained in the methods section. The error bars indicating 95% confidence intervals from the mean values for inactivation efficiency in Fig. 2a–c are shown in Supplementary Figs. 2–4 in the SI. The red dashed line represents the lethal electroporation threshold (LET) where overall inactivation reaches 50%. (**d-f**)

The overall inactivation percentage over increasing Cu concentration for increasing values of electric field strength and their corresponding linear fit lines for pulse widths 500 ns (**d**), 1 μs (**e**), and 2 μs (**f**). The slope values for linear fit lines shown in panels **d**–**f** plotted for pulse widths 500 ns (**g**), 1 μs (**h**), and 2 μs (**i**). There is no statistical analysis provided for Fig. 2d–i as they are data analyzed and obtained from the data points, trends, and relationships shown in Fig. 2a–c and not pulled from the total replicate data.

suggests a combination effect of the two approaches can increase the efficacy of treatment. Additionally, for all three figures there is a clear increase observed in the total inactivation values as the Cu concentration is increased, which was expected. For longer pulse widths, there are slightly lower electric field strengths applied to avoid reactive oxygen species (ROS) generation and bubble formation (39 kV/cm for 500 ns pulse width, 32 kV/cm for 1 μs pulse width, and 25 kV/cm for 2 μs pulse width). Despite this limitation, there is still an observable higher inactivation percentage with a longer pulse width applied. When looking at all three figures for 2 mg/L Cu (purple star curves), 25 kV/cm applied at 2 μs pulse width (Fig. 2c) achieves ~90% inactivation, while 25 kV/cm at 1 μs (Fig. 2b) and 500 ns pulse width (Fig. 2a) achieve only ~35%. Similarly, 30 kV/cm applied at 1 μs pulse width (Fig. 2b) achieves ~84% inactivation, while 500 ns pulse width (Fig. 2a) only achieves ~43%.

**Synergistic effect**

Figure 2d–f shows the overall inactivation percentage for increasing Cu concentration for a higher range of electric field strengths. The bottom lines (grey hexagon curve) represent the conditions with no

EFT and the consecutive lines above it are for a few of the electric field strength values applied. Although the combination approach clearly indicates the enhancement of inactivation when using the two methods at once, it does not directly conclude the synergistic effect. To investigate whether EFT and Cu can have a synergistic effect, the linear slope values are calculated for various electric field strengths and summarized in Fig. 2g–i. These slope values represent the Cu dose response, i.e., % inactivated per Cu concentration dosed in mg/L. For all pulse widths, the slope for 0 kV/cm indicates that without any EFT present, the device can achieve ~16% inactivation efficiency per mg/L of Cu dosed. With application of the EFT at a pulse width of 500 ns (Fig. 2g), EFT-Cu's synergistic dose response increases to ~27% inactivation for 32 kV/cm, and ~38% inactivation for 35 kV/cm applied, concluding that with increasing electric field strength applied, the relative synergistic dose responses increase. Similar analysis was conducted for 1 and 2 μs pulse widths as it was hypothesized that increasing pulse width (duration of each pulse), increases the duration a pore may be open on the cell surface, potentially leading to more Cu uptake and a higher inactivation efficiency[39]. For example, when looking at 500 ns pulse width

(Fig. 2g), a slope of ~27% is achieved with 32 kV/cm applied, while for 1 µs pulse width (Fig. 2h) a slope of ~29% is achieved with 30 kV/cm, and for 2 µs pulse width (Fig. 2i) an ~29% slope is achieved with only 25 kV/cm applied. As we increase pulse width, the synergistic dose response for EFT-Cu inactivation efficiency increases, allowing us to use lower electric field strengths to obtain the same slope values. This observation allows for the conclusion of the synergistic effect of EFT-Cu at the microscale.

### Operando observation and synergistic effect of EFT-Cu over time

To study the rate of inactivation by EFT and/or Cu, the LOAC setup was observed in situ for 180 min, using operating parameters for 500 ns pulse width, for three separate conditions: (1) Cu-only (2 mg/L), (2) EFT-only (0 mg/L), and (3) EFT-Cu (2 mg/L). The fluorescent images for the center portion of the device channels during these three treatment conditions over time are shown in Fig. 3a–c, respectively. Overall inactivation here refers to positive PI-staining further explained in Supplementary Note 5. For Cu-only (Fig. 3a), there is an increase of fluorescence (bright red dots) as the number of stained and dead cells increases over time. EFT-only (Fig. 3b) results in quick and high inactivation at the center, where the electric field strength is at its strongest. Beyond this, there is no significant increase of staining or fluorescent cells over time. EFT induced inactivation can be assumed to be a one-time process, rapid and with no residual effect in this case. For EFT-Cu, the observation over three hours reveals what we expected. As shown in Fig. 3c, the EFT induced inactivation is present predominantly at the center regions, while the Cu induced deaths occur at any point along the channel. This displays the successful combination of EFT-Cu in situ and over time.

The time series images shown in Fig. 3a–c were further analyzed for each condition in order to quantify the inactivation percentage at every electric field strength for every time point. This is represented in Fig. 3d, the inactivation percentage over time for the three conditions for six different electric field strengths. Each individual figure shows an almost linear curve of inactivation for Cu-only (gold downward triangle), a nearly flat line for EFT-only (grey square), a dashed empty triangle line to represent the calculated theoretical additive of the combination of EFT and Cu (done by adding each consecutive Cu-only and EFT-only points together), and a final curve for EFT-Cu (blue solid triangle), representing the measured combination results (additional plots shown in Supplementary Fig. 5). Not only is there a clear enhancement when combining the two approaches, but it also indicates the natural residual effect of Cu can be further taken advantage of with the addition of the EFT, and a faster overall disinfection performance can be achieved. Specifically for the electric fields strengths 35 & 37 kV/cm, the theoretical additive reaches 50% inactivation at ~165 and ~100 min, while the measured EFT-Cu results achieve the same at ~35 and ~15 min. This result suggests that at higher electric fields strengths, the synergistic effect of EFT-Cu can lead to inactivation efficiencies at a rate five times greater than predicted. In addition, for 33 kV/cm at the 90 min mark, the theoretical additive achieves only ~23% inactivation while the EFT-Cu already measured ~77%, resulting in a calculated synergistic enhancement for inactivation to be up to 300% higher than if EFT and Cu were applied separately. Lastly, the numerous plots shown in Fig. 3d (along with the additional plots in Supplementary Fig. 5) display varying results depending on the strength of the electric field and the time for inactivation. Our previous results suggest this data would also fluctuate with a changing Cu concentration, strongly suggesting the synergistic effect can be customized for various applications to achieve high disinfection performance while optimizing for either a lower Cu concentration, shorter treatment time, or lower energy consumption.

### Single cell analysis of EFT-Cu

In this section, the focus was on observing the membrane permeability in a singular cell in each of the three conditions operando. To visually compare, Fig. 4a–c shows the images of a single cell in each of the three conditions. The cells either show fluorescence from Cu ion permeation alone (Fig. 4a), electroporation damage through EFT (Fig. 4b), or the combination effect where EFT is applied first, and damage to the cell occurs from Cu ion permeation after pulses are removed (Fig. 4c) (short videos for each condition are available in Supplementary Movies 1–6). The quantified intensity of dye saturation for each image is also normalized over time as shown in Fig. 4d–f along with another individual cell showing similar results. Figure 4a shows the dye staining for Cu-only, where time 0 represents the moment before any fluorescent intensity is observed or measured, and full saturation (intensity 100%) is measured and observed at ~44 s. Figure 4d further confirms a slow and steady increase from Cu ion permeation for Cu-only. In Fig. 4b for EFT-only, the cell shown in the images is exposed to a high electric field strength region and quickly damaged through electroporation. According to our results from the previous sections, the EFT-only condition can achieve ~40% overall inactivation, while the other ~60% of cells continue to observe no damage. The single cell shown in Fig. 4b is specifically one of the ~40% of cells most likely inactivated through EFT alone. It is important to note that the cell is infiltrated with dye rapidly here and measures complete saturation at only ~3.3 s as indicated by the non-linear trend in Fig. 4e. In Fig. 4c for EFT-Cu, the cell is exposed to EFT while dosed with Cu ions, but not observed to be fully saturated with dye until several minutes after pulses are removed. Different to the EFT-only case, some of the other ~60% of cells still intact after pulses are removed, are observed to be damaged here later on by the Cu ions in solution using the combination approach of EFT and Cu. The image frames for EFT-Cu show the cell is saturated with dye at a faster pace than for Cu-only as the cell measures and observes complete saturation within ~21 s, or less than half the time. EFT-Cu shows a measured trend (Fig. 4f) comparative to the combination of EFT-only and Cu-only, a slow and steady start with a sharp increase to reach complete saturation. This result suggests there is an observable difference in the rate the cell membranes are being damaged using these two methods independently versus when they are combined.

## Discussion

From our understanding of EFT and Cu's independent mechanisms to inactivate bacteria and the observed results of the single cell study, there is further evidence pointing to an increased cell permeability as a result of the applied EFT. For Cu-only (Fig. 4a), the cells show successful membrane damage with PI staining, but the process to achieve complete staining takes long (~44 s). This is concurrent with literature on copper induced contact killing of microorganisms, as it may take minutes to hours for inactivation to occur (Fig. 3a for Cu only inactivation ranges from 1 min to 3 h)[40,41]. Studies have correlated this cell damage and death with lipid peroxidation, loss of membrane integrity, and Cu ion uptake[42–44]. These studies also agree that DNA degradation is not the primary cause of cell death by Cu, but rather the gradual release of genetic material from membrane damage[43,44]. For EFT-only (Fig. 4b), this process occurs in under 10 s (~3.3 s) for higher electric field strength regions. When the electric field strength is strong enough and applied for long enough, it will cause damage to the cell membrane or create pores much faster, allowing for rapid diffusion of the dye and quicker inactivation as shown[24]. After pulses are removed in EFT-only, we observe PI staining to continue for a very short period (few seconds). This is concurrent with literature as Li et al. confirmed diffusion of PI into cells continue after pulses are ceased with fluorescence intensity increasing shortly after pulses were removed[45]. When the combination approach EFT-Cu is applied, there are some cells that stain rapidly from the EFT cell damage, and numerous others that continue to stain over the next few minutes to hours. After the pulses

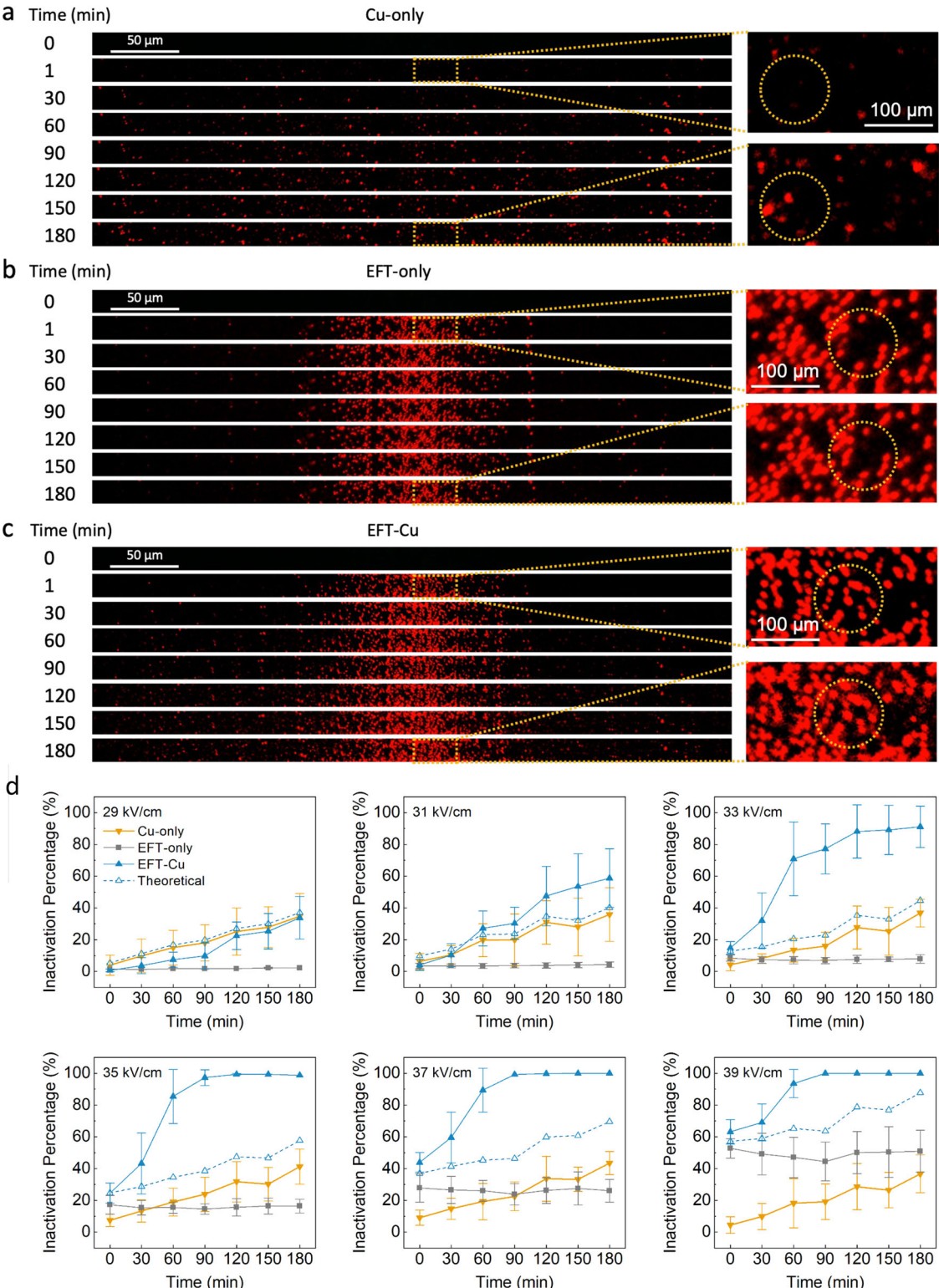

**Fig. 3 | Microscopy images of the LOAC device under fluorescent channel and quantified inactivation percentage taken every 30 min over the course of 3 h.** Cu-only using no EFT + 2 mg/L Cu (**a**), EFT-only using EFT + 0 mg/L Cu (**b**), and EFT-Cu where EFT + 2 mg/L Cu (**c**) are all shown. The pulse width for EFT is operated under 500 ns. Images for 1 min into the experiment are also presented for each condition to show cell damage immediately following any pulse application. The overall inactivation here refers to positive PI-staining further explained in Supplementary Note 5. Due to time and processing constraints, there are only 5 replicate channels for each experiment collected and analyzed in the time series results presented (compared to the usual capacity of 30). Only one representative channel for each experiment is shown. The zoomed in portions are shown for 1 and 180 min for each condition, and areas of significance are highlighted by yellow dashed circles to guide the readers. **d** The overall inactivation percentage over time in minutes for Cu-only, EFT-only, and EFT-Cu. The dashed line refers to the calculated theoretical additive. Individual plots are shown for specific electric field strengths appearing in increasing order from left to right and top down. The error bars represent 95% confidence intervals from the mean values for the 5 total replicates collected and analyzed for each of these and all additional plots with electric field strengths 29-39 kV/cm. (For any references to colors in the figures the readers are referred to the online/web version of this article).

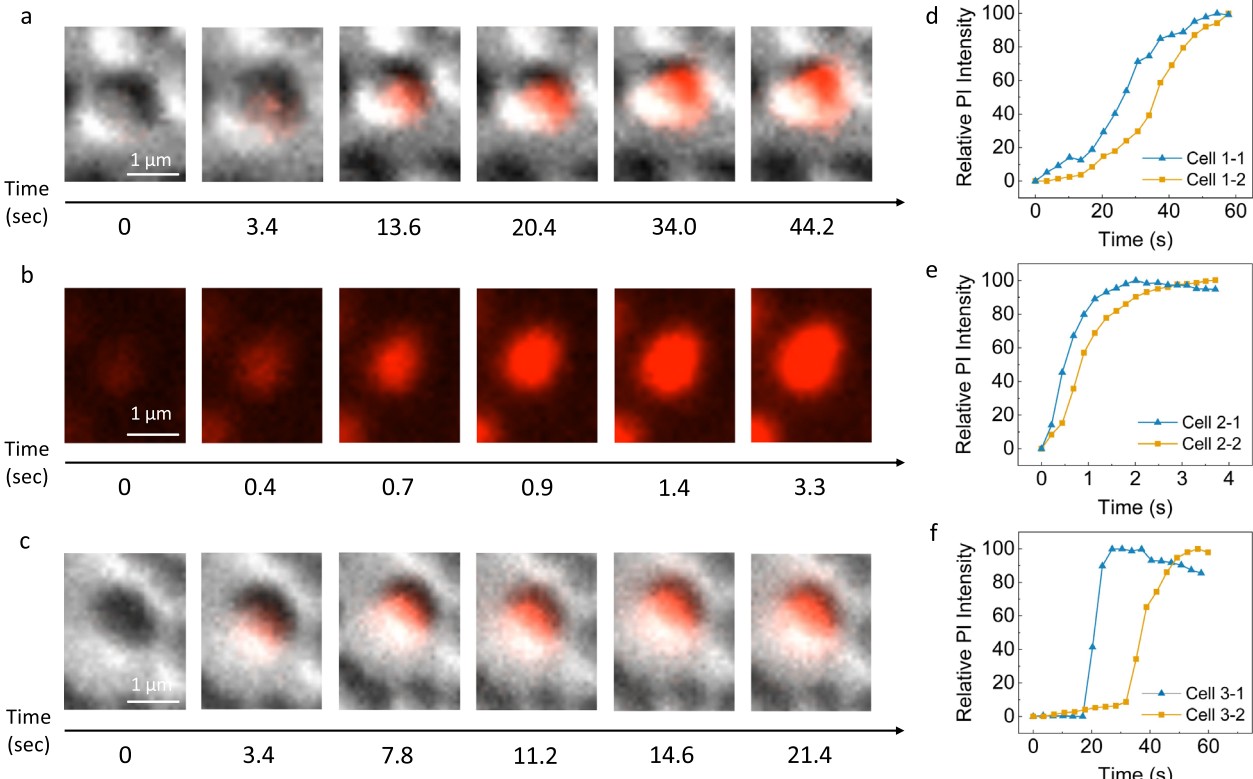

**Fig. 4 | Microscopy images of the LOAC device under fluorescent and DIC channels where the red indicates PI fluorescence and diffusion of the dye into the individual cells and the relative time points where each image's time stamp is shown underneath.** Cu-only using no EFT + 2 mg/L Cu (**a**), EFT-only using EFT + 0 mg/L Cu (**b**), and EFT-Cu where EFT + 2 mg/L Cu (**c**) are all shown. Conditions for Cu-only and EFT-Cu show both the DIC and fluorescent channels, while the EFT-only condition shows only the fluorescent channel due to time and processing constraints. Short videos for each condition are available in Supplementary Movies 1–6. The pulse width for EFT is operated under 2 µs. (**d**–**f**) The relative PI intensity of fluorescence as measured by the microscope normalized over time in seconds is also shown for Cu-only (**d**), EFT-only (**e**), and EFT-Cu (**f**) where importance is placed on when the cells reach 100% saturation of the dye intensity. (For any references to colors in the figures the reader is referred to the online/web version of this article).

are applied, these cells are achieving complete staining at a rate faster than that of only Cu ion disinfection (Fig. 4c), indicating the application of the EFT is damaging or weakening the membrane for cells that have not yet been damaged from electroporation alone. Since exposing bacteria to external electric fields can alter their structural integrity, this result suggests EFT is increasing the permeability of the cell membrane for Cu ions to act more quickly in creating more permanent damage to the cells and inevitably cause cell death[10,21,25]. Not only is the diffusion of the dye faster, but also the number of bacteria that continue to fluoresce over time increases (confirmed in earlier sections). We are confident the combined approach has an enhancement effect leading bacteria to be weakened by the EFT and more susceptible to inactivation by Cu ion permeation. This observation allows us to visually conclude the synergistic effects of EFT-Cu at microscale.

For extended perspective on this work, the LET (lethal electroporation threshold for 50% of bacteria inactivation), was also collectively determined for the three pulse widths and five Cu concentrations tested (results shown in Supplementary Fig. 6). As expected, the LET decreased with the increase of copper concentration for all cases of varying pulse widths applied and all decreased as pulse widths increased[39]. Both the LET and previous results strongly suggest that the synergistic effect can be customized to achieve a tradeoff between having lower Cu concentration, shorter treatment time, or lower energy consumption. These findings can now be used to optimize and customize the disinfection performance of EFT-Cu to adapt for its various applications. Future studies can look to optimize this for either faster disinfection performance using higher Cu concentrations/energy requirements, or minimal chemical/energy consumption using lower Cu concentrations and the optimal electric field strength.

The LOAC device developed and used in this study drastically enhances our ability to perform electric pulse experiments for various values to gather large amounts of data. We have continued to improve and re-develop our devices, but still, it does not come without setbacks. As we focus mainly on understanding the mechanism, the device does a great job of immobilizing the cells for optimal observation and imaging. Despite this, it is important in future work to configure a way to translate this study from the microscale to bench scale. Future work should involve developing new methods to observe flow for a plug flow reactor at microscale to reduce the limitations that come with immobilizing bacteria and best mimic the reactor conditions for real-life applications. This will need to account for the electric field strength applied per distance of separation from the bulk electrode, the need for an alternative power source to apply higher voltages, and a new method of analyzing the similar synergistic effects. We are confident, when translated to a larger, more realistic system, the trends of synergistic effects observed in this study will apply to other systems at greater scale. Another limitation of using the LOAC platform includes the inability to perform standard plate count experiments with the tested bacteria. Although we rely heavily on the established method of cell staining with propidium iodide, the too few and well immobilized cells on our device make extraction for plate count very difficult. In more traditional batch experiments of larger volumes and bacteria concentration, the plate count method could be used for analysis. The limitation of these batch experiments is the inability to test a range of electric field strength in one experiment and

receive detailed results and images such as those obtained using the LOAC device in this study. Other limitations of this study stem from the generation of ROS at higher electric field strength conditions. Because we focused strictly on studying EFT and Cu effects only, a wider range of parameters inclusive to ROS were not able to be studied. There is great potential for future work on the combination approaches of EFT with ROS. Lastly, EFT-Cu has been studied at the microscale in depth in this work, but there are still many unknown variables regarding the synergy of these two approaches. Future work can test other conditions and parameters as reviewed in more traditional pulsed electric field experiments, *e.g.*, a wider range of pulse widths, different total effective treatment times, and effects of conductivity, pH, temperature, and turbidity of the liquid[46]. Finally, great importance should be placed on studying the potential enhanced and synergistic effects of combining EFT with other established disinfectants and methods of disinfection as great promise is shown for the combination of both physical and chemical based methods using EFT-Cu.

## Methods

### Fabrication of the Lab-on-a-Chip Device

The Lab-on-a-Chip (LOAC) device (Fig. 1a) was constructed by depositing gold electrodes onto a glass substrate using standard photolithography and lift-off methods (Fig. 1b)[39]. First, a layer of negative photoresist (PR) (Futurrex NR9-1500py) was spin-coated onto the surface of a 4-inch glass wafer (Schott Borofloat 33). The wafer was then exposed to UV light to formulate the desired pattern of electrodes using a maskless aligner (Heidelberg MLA 150). After this step, the pattern was developed using resist developer (Futurrex RD-6). Following this, 20 nm thick titanium (adhesion layer) and 200 nm thick gold layers were deposited onto the glass surface using e-beam evaporation (Denton Explorer 14–E-beam Evaporator)[47]. Any residual PR was removed using acetone (Transene UN1090 Acetone, LM Grade, 3, PG2) and all procedures followed general best practices of metal deposition and lift-off processes. Zoom-in microscopy images of the channels are also shown. Each chip was designed to have 30 repeating units for conducting parallel experiments under each condition. A symmetric electrode design with curved edge was used to generate a range of electric field strengths that change linearly along the x-axis (Fig. 1c, d, further detailed in Supplementary Note 2).

### Cell preparation and immobilization

*Staphylococcus epidermidis* (*S. epidermidis*) (ATCC, cat# 12228) was used as the model bacterial strain to quantitatively assess the synergistic effect between EFT and Cu as it has been traditionally used in experimentation for water disinfection and pulsed electric field research[46,48]. The bacteria culture was prepared using a standard method, incubation in nutrient broth (Becton Dickinson, cat# 234000) at 35 °C for ~15 hr. An aliquot of 1 mL of cell culture was then centrifuged at 1000 *g* for 5 min, and the pellet was resuspended with 10 mM phosphate buffer solution (pH 8.5). This process was repeated 3 times to obtain the solution consistently used for our experimentation.

To conduct consistent and precise experiments with an applied electric field, the bacteria need to be immobilized to the surface of the LOAC. For this reason, the chips were pre-coated with positively charged poly-L-lysine (Sigma-Aldrich, cat# A-005-C) so that the model bacteria (negatively charged) could be uniformly distributed and securely attached onto the surface (pre-coating method further detailed in Supplementary Note 3). The prepared bacterial solution was then deposited onto the surface of the chip using roughly 50–100 µL of solution and left to settle and immobilize to the surface of the LOAC for 50 min. The chip surface was then gently rinsed with DI water to remove the excess bacterial solution and any remaining unattached cells, leaving the attached bacteria ready for experimentation.

### EFT-Cu treatment

Cu ion solutions were prepared by dissolving copper sulfate (Alfa Aesar, cat# 33308-36) in DI water. For all experiments, concentrations of 0–2 mg/L as Cu were used. As the conductivity of solution can alter the strength of electric field applied and the uptake of PI into the cell, all Cu solutions were made with the addition of sodium sulfate (VWR, cat# 97062-438) to ensure the conductivity of all solutions resulted were the same as that of the 2 mg/L solution (approximately ~10 µS/cm) for consistency. The conductivity was measured using an Orion Versa Star Pro conductivity probe (Thermo Scientific). Square-wave pulsed voltage shocks (50–80 V) with different parameters, (*i.e.*, pulse width, period, and pulse number), were applied. Nevertheless, the effective treatment time (*i.e.*, product of pulse width and pulse number) was kept at 20 ms and the duty cycle (*i.e.*, ratio of pulse width over period) was kept at 0.1% during all conditions (schematic of EFT terms shown in Supplementary Fig. 7). The three different EFT conditions included: (1) a pulse width of 500 ns, a period of 500 µs, and a pulse number of 40k; (2) a pulse width of 1 µs, a period of 1 ms, and a pulse number of 20 k; and (3) a pulse width of 2 µs, a period of 2 ms, and a pulse number of 10k. The pulses were applied to the LOAC device using a high-speed pulse generator (Avtech AV-1010-B) triggered with a waveform generator (Keysight 33509B) (further details on parameter setup, procedure of pulse applications, and avoiding heating effects, bubble, and ROS formation are discussed in Supplementary Note 4). For all experiments, after Cu solution and/or electric field was applied, there was a 2-hour residual effective treatment time (unless otherwise stated) to best collect the most representative results at this scale.

### Imaging and data processing

For the bulk of the experiments, after Cu solution and/or electric field was applied, there was a 2-hour effective treatment time and rest period after pulses were removed to best collect the most representative results at this scale. From our preliminary studies, we found that there was no significant difference in the results obtained using a single staining or double staining method and a single staining method would also result fewer potential inaccuracies, and thus, a single staining method was used. Most of the data analysis was done through quantifying the number/percentage of cells inactivated, over time, and in specific electric field regions. For all cases, the results were indicated using standard, well-established propidium iodide (PI (Invitrogen), 2 µM) staining. PI molecules will only enter bacteria with a compromised membrane and bind to their DNA to result a fluorescence enhanced several folds as reiterated by many other studies on electroporation and membrane permeability[45,46,49]. In case any reversible electroporation was present from the pulse application, the cells were only stained at the end of the experiment, after treatment, the 2 h rest period, and right before imaging. Since any reversible membrane damages should be recovered after the 2-hour period, we consider the cells stained with PI as inactivated for the bulk of the results presented (Fig. 2)[50,51]. The results presented in Figs. 3 & 4 regarding the time series and single cell analysis have a slightly different methodology and are further detailed in the Supplementary Note 5.

The microscope images captured were through differential interference contrast (DIC) and a fluorescent channel using a Zeiss inverted fluorescent microscope (Axio observer 7) connected to a CCD camera. MATLAB (version 2023a, MathWorks) was used to process the images and calculate the inactivation efficiency. The results from all viable channels out of a total 30 repeating units on one chip were averaged and used collectively to determine the inactivation percentage in all cases. With the symmetry of the channels, there are a total of 120 data points (60 values, each with a duplicate) collected for the electric field strengths resulted from each individual channel. Further details on the code analysis are discussed in Supplementary Note 5. Supplementary Code 1 is also provided with this paper.

## Statistics and reproducibility

As this study is complex, it was designed specifically with consistency and robustness in mind. Most results have a wide range of data to be analyzed as the LOAC produces a total of 30 replicate experiments on one singular device. This results large quantities of data for each individual condition tested in our study. All collected data in this study came from images collected under the fluorescence microscope. All collected microscope images were then analyzed consistently using the MATLAB script developed for this study. No statistical method was used to predetermine the sample size. Some data were excluded from the analysis. Specifically, any individual channels that showed damage from wear and tear of the device usage or poor/inconsistent connectivity with the bulk electrode. The fabrication process and experimental procedure all involve inadvertent human error, because of this, typically 5–10 individual channels could be excluded from a total of the 30 replicates for any condition. The experiments were not randomized, and the investigators were not blinded to allocation during the experiments and outcome assessment. The statistical analysis for the bulk of the results presented were calculated as 95% confidence intervals for all collected replicate channels out of the total 30.

## Reporting summary

Further information on research design is available in the Nature Portfolio Reporting Summary linked to this article.

## Data availability

All data supporting the findings of this study are available within the article and its supplementary files. Any additional requests for information can be directed to, and will be fulfilled by, the corresponding authors. Source data are provided with this paper.

## Code availability

The scripts for the data analysis were developed in MATLAB (version 2023a, MathWorks) and are provided with this paper as Supplementary Code 1.

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

## Acknowledgements

The authors acknowledge funding from the National Science Foundation (grant number CBET 1845354, X.X.). This work was performed in part at the Georgia Tech Institute for Electronics and Nanotechnology, a member of the National Nanotechnology Coordinated Infrastructure (NNCI), which is supported by the National Science Foundation (grant number ECCS-2025462, M.J.).

## Author contributions

M.J., T.W., and X.X. designed the research. M.J. performed the research experiments. M.J., T.W., and X.X. analyzed the data and wrote the paper.

## Competing interests

The authors declare no competing interests.
