## [Peer Review File · Nature Communications]

REVIEWER COMMENTS

Reviewer #1 (Remarks to the Author):

The investigation of synergistic effects of PEF (EFT here) and copper ions were investigated. The work itself is worth being published, but I am not sure is it the right journal. More specific journals can attract more potential readers. Nevertheless, the manuscript is well-written and mature for publishing. However, from this point of view, there are serious concerns as below.

- 1) The critical experiment is missed. Inactivation here means PI positive. As it was shown here in Figure 4, the PEF increases the permeability of the plasma membrane the viability (plate count or similar) experiment of the bacteria cells is critical.
- 2) Single Cell Analysis of EFT-Cu. Figure 4. It looks like the EFT alone works much (10x) better than EFT-Cu in combination. Check the figure note on lines 270-271.

Reviewer #2 (Remarks to the Author):

The ability to address antibiotic resistant microorganisms is critical and pulsed electric fields (PEFs) combined synergistically with other mechanisms is an important way to accomplish this [see review paper included in the comments below -- Garner AL. Pulsed electric field inactivation of microorganisms: from fundamental biophysics to synergistic treatments. *Applied microbiology and biotechnology*. 2019 Oct;103(19):7917-29.]. The idea to use copper in combination with PEFs is interesting and will be an important step toward developing an important approach in water. While this is an important result, review articles and many other papers demonstrating synergistic applications of PEFs for microorganism inactivation and cancer treatment (with drugs, heat, UV, and natural products, just to name a few) make it difficult for me to see this as a *Nature Communications* paper. Since it does include some mechanistic information that expands upon just synergistic effects, and it addresses a critical problem, I think it could be an excellent fit in another Nature family journal, particularly *Nature Water* or *Communications Biology*.

Before even considering that, I note below the absence of error bars and statistical analysis, which are critical when considering biological phenomena such as these and must be included in figure captions and in the text. How many replicates were run? What were the standard deviations? What sort of statistical analyses were run?

Overall, this is excellent and detailed work, but the results require more statistical analysis and I need further convincing that this is the correct venue.

1. Fig. 1g: There are no error bars. How many replicates were run? Is there a standard deviation or error that can be reported?
2. Fig. 2: Again, error bars and number of replicates?
3. It is very unusual in bioelectrics experiments to see such fine granularity in results as shown in Fig. 2 – just changing the electric field by less than 5% leads to vastly different inactivation efficacies. Usually, statistical differences obscure this sort of fine detail. Please discuss.
4. Fig. 3 – statistical analysis?
5. Another mechanism to consider (or at least mention) is electrophoresis. See this paper for an example: Li J, Lin H. Numerical simulation of molecular uptake via electroporation. *Bioelectrochemistry*. 2011 Aug 1;82(1):10-21.
6. Consider citing this manuscript outlining synergistic treatments of microorganisms with pulsed electric fields as it seems particularly relevant to the last paragraph of the manuscript: Garner AL. Pulsed electric field inactivation of microorganisms: from fundamental biophysics to synergistic treatments. *Applied microbiology and biotechnology*. 2019 Oct;103(19):7917-29.
7. With 40,000 pulses applied, how was sample heating controlled? Heating can have a significant effect on inactivation, and it is important to at least qualitatively understand the relative contribution of heating and electric field effects.

Operando Investigation of the Synergistic Effect of Electric Field Treatment and Copper for Bacteria Inactivation

Mourin Jarin¹, Ting Wang¹, Xing Xie^{1,2*}

¹School of Civil and Environmental Engineering, Georgia Institute of Technology, Atlanta, Georgia 30332, United States

²Institute for Electronics and Nanotechnology, Georgia Institute of Technology, Atlanta, Georgia, 30332, United States

*Corresponding author: Xing Xie, Email: xing.xie@ce.gatech.edu, Phone: 4048949723

Response to reviewers' comments:

We thank the reviewers for the constructive and helpful comments. Their insights and feedback have allowed us to improve the manuscript. Below, we have addressed the comments and concerns raised by each reviewer in detail.

Response to Reviewer #1

The investigation of synergistic effects of PEF (EFT here) and copper ions were investigated. The work itself is worth being published, but I am not sure is it the right journal. More specific journals can attract more potential readers. Nevertheless, the manuscript is well-written and mature for publishing. However, from this point of view, there are serious concerns as below.

We appreciate the reviewer's overall assessment and recognition of our research and findings. We additionally want to thank the reviewer for pointing out some areas of concern that we can improve upon. The following breaks down each of the reviewer's two comments in detail.

1. The critical experiment is missed. Inactivation here means PI positive. As it was shown here in Figure 4, the PEF increases the permeability of the plasma membrane the viability (plate count or similar) experiment of the bacteria cells is critical.

The reviewer is correct that "inactivation" means PI positive in this work. Live/dead cell distinguishing probes such as Propidium Iodide (PI) are widely used to evaluate cell inactivation for microscopy, flow cytometry, and microplate formats. PI molecules will only enter dead bacteria with a compromised membrane and bind to their DNA to result in a fluorescence enhanced several folds as reiterated by many other studies on electroporation and cell death.^{1,2}

We appreciate the reviewer's critical feedback on the need for a plate count or culture of the final cells in order to confirm cell viability. We agree that for traditional biological experiments this is a critical and important factor in reassuring the confidence of results. However, due to the setup of our experiment, where the cells must be immobilized to the surface of the chip, it is extremely challenging to remove the bacteria from the electrode surfaces without harm and then plate them. Another limitation is the small surface area and therefore low number of cells to begin with. Although great for imaging, it is difficult to remove and retain a quantifiable number of bacteria from the chips to do a standard plate count test.

We appreciate the comment as we have considered this limitation in our present and past work using lab-on-a-chip platforms. In this and other previous studies, there have been concerns regarding reversible electroporation and the false positives it may introduce to our inactivation results. Because of this concern, we perform preliminary experiments where the pulses are applied and hours later, we will check the cells with PI staining in order to confirm the results are similar to those stained before, during, or after the pulses are applied. The conditions and pulse parameters we have used in this study also provide increased confidence in limiting false positives with inactivation and is also why we have stuck with consistently using standard and established PI staining as our alternative to plate count experiments in previous and current research involving lab-on-a-chip devices.³⁻⁵

To address the reviewer's concern, we have added an additional portion of discussion for recognizing the limitation of the lab-on-a-chip device and its inability to perform standard plate count experiments in the final paragraph. There are slight edits made to the methods section specific to imaging and data processing to reiterate the use of propidium iodide as our confident

method of cell viability indication. The revisions to the manuscript are as follows and highlighted in the manuscript document as well.

Line 336: Another limitation of using the LOAC platform includes the inability to perform standard plate count experiments with the tested bacteria. Although we rely heavily on the established method of cell staining with propidium iodide, the too few and well immobilized cells on our device make extraction for plate count very difficult. In more traditional batch experiments of larger volumes and bacteria concentration, the plate count method could be used for analysis. The limitation of these batch experiments is the inability to test a range of electric field strength in one experiment and receive detailed results and images such as those obtained using the LOAC device in this study.

Line 404: For all cases, the cell viability was indicated using standard, well-established propidium iodide (PI, 2 mM) staining. PI molecules will only enter dead bacteria with a compromised membrane and bind to their DNA to result a fluorescence enhanced several folds as reiterated by many other studies on electroporation and cell death.^{1,2}

2. Single Cell Analysis of EFT-Cu. Figure 4. It looks like the EFT alone works much (10x) better than EFT-Cu in combination. Check the figure note on lines 270-271.

We appreciate the reviewer's comment on the result displayed in Figure 4. We agree that it was not represented for the best understanding of the readers. Apologies for the confusion. EFT-only here does work extremely quickly to electroporate and inactivate the single bacteria cell shown. It is noted shortly in the text, but for further clarification, the pulsed electric field is capable of killing some bacteria almost instantly and other not at all. For example, in the condition using 2 ms pulse width, the electric field strength ranges from 5-25 kV/cm and the highest cell death recorded by EFT-only is ~40%. With no Cu ions, the EFT-only condition has no effect on ~60% of bacteria. The result shown in Figure 4b for EFT-only is of the 40% killed by electroporation immediately when the pulsed electric field is applied. There are subsequently many other bacterial cells that are not inactivated by only this method that we do not show because it is not interesting to observe. What we show in the following Figure 4c is the death of a single cell using EFT and Cu. As we add Cu ions into the experiments, the inactivation efficiency increases. In Figure 4c, the cell we focus on was also exposed to the same pulsed electric field, in a very similar region as Figure 4b. This cell was not initially killed by the EFT (the other 60% for example) but inactivated later on with the presence of Cu ions. What we find in experiments with EFT-only are that the EFT can kill bacteria very quickly while the pulses are applied, but the inactivation doesn't continue after pulses are removed (Figure 3b). Although it is a one-time treatment, the addition of Cu ions in Figure 4c elongates the inactivation for several hours (also shown in Figure 3c), which EFT-only cannot perform the same.

To address the reviewer's concern, we have added an additional portion of discussion in the results addressing Figure 4 on the details of what is shown specifically for EFT-only for clarification to the readers. The revisions to the manuscript are as follows and highlighted in the manuscript document as well.

Line 261: In **Fig. 4b** for EFT-only, the cell shown in the images is exposed to a high electric field strength region and quickly killed through electroporation. The EFT-only condition here can achieve ~40% overall inactivation, while the other ~60% of cells continue to observe no damage. The single cell shown in **Fig. 4b** is specifically one of the ~40% of cells killed through EFT alone. It is important to note that the cell is infiltrated with dye rapidly here and measures complete saturation at only ~3.3 s as indicated by the non-linear trend in **Fig. 4e**. In **Fig. 4c** for EFT-Cu, the cell is exposed to EFT while dosed with Cu ions, but not observed to be killed until several minutes after pulses are removed. Different to the EFT-only case, some of the other ~60% of cells still intact after pulses are removed, are observed to be inactivated here later on by the Cu ions in solution using the combination approach of EFT and Cu.

Response to Reviewer #2

The ability to address antibiotic resistant microorganisms is critical and pulsed electric fields (PEFs) combined synergistically with other mechanisms is an important way to accomplish this [see review paper included in the comments below -- Garner AL. Pulsed electric field inactivation of microorganisms: from fundamental biophysics to synergistic treatments. *Applied microbiology and biotechnology*. 2019 Oct;103(19):7917-29.]. The idea to use copper in combination with PEFs is interesting and will be an important step toward developing an important approach in water. While this is an important result, review articles and many other papers demonstrating synergistic applications of PEFs for microorganism inactivation and cancer treatment (with drugs, heat, UV, and natural products, just to name a few) make it difficult for me to see this as a *Nature Communications* paper. Since it does include some mechanistic information that expands upon just synergistic effects, and it addresses a critical problem, I think it could be an excellent fit in another *Nature* family journal, particularly *Nature Water* or *Communications Biology*.

Before even considering that, I note below the absence of error bars and statistical analysis, which are critical when considering biological phenomena such as these and must be included in figure captions and in the text. How many replicates were run? What were the standard deviations? What sort of statistical analyses were run?

Overall, this is excellent and detailed work, but the results require more statistical analysis and I need further convincing that this is the correct venue.

We appreciate the reviewer's overall assessment and recognition of our research and findings. We additionally want to thank the reviewer for pointing out some areas of concern that we can improve upon. Additional thanks for providing us with a review paper very relevant to our future studies related to synergistic mechanisms with EFT. The review paper by Garner AL is further discussed in response to comment 6 below.

First, to address the reviewer's concerns about whether *Nature Communication* is a good fit for our article, we believe that it is due to the interdisciplinary nature of the work presented. The work we present on EFT and Cu is targeted towards a broader audience of readers than that of *Nature Water and Communications Biology*. There are various applications for the results shown and are not limited to only fields like water treatment and liquid/food processing, but also to potential antimicrobial surfaces in the future. The experimental method applied in this study, operando investigation of the microbial inactivation processes using LOAC devices, can also be applied in my other studies in relevant fields. Because the scope of the results can interest a broad range of readers, we believe *Nature Communications* will be an optimal journal for publication and dissemination of our research.

Lastly, to shortly summarize the reviewer's main concerns for statistical analysis in our work, we have included several detailed responses to explain our statistical analysis and reintroduced many of the results with their statistical error presented. Because some of the results with error bars may be too dense and confusing in the main manuscript, they have been proportionately added to the *Supplementary Information*. The following breaks down each of the reviewer's seven comments in detail.

1. Fig. 1g: There are no error bars. How many replicates were run? Is there a standard deviation or error that can be reported?

We appreciate the reviewer's feedback on the need for error and statistical analysis as we agree it is very important in standard biological experiments. The beauty of our lab-on-a-chip device is specifically the design to achieve a range of continuous electric field strengths with one applied voltage, while also having up to 30 replicate channels on every individual chip. To further clarify, the electrode was designed purposefully to be curved for continuous change of the electric field strengths. This results in almost infinite data obtainable, with the analysis process determined by the size of the cells and the number of sections we break each channel into. In our analysis, we choose to section the range of electric field strength values into 120 columns. Please refer to Figure 1a (zoomed in singular channel with red dashed rectangle) for reference to the analysis portion we section into 120 columns. With the symmetry of the channels, there are a total of 120 data points (60 values, each with a duplicate) of electric field strengths resulted from each individual channel in an EFT experiment. This is true for every channel and the replicated experiments are all on the same device, as all viable repeating channels are analyzed for each data point represented. For most cases, like in Figure 1g, there are 30 replicates of each channel, resulting in 60 total data points for each electric field strength that are averaged together to result each individual data point the data shown to the readers in Figure 1g.

Using this device, breaking down the channel into 120 columns for analysis, we obtain very strong trends, along with some error in the inactivation data. If we were to use a smaller number of columns, e.g., 10, there would be much less error in the data, but also fewer electric field strength values and a less convincing and detailed result. We are confident in the conclusion supported by our analysis, as the phenomenon is verified under three conditions with different pulse widths.

We apologize for not making this clearer in our manuscript and agree that it is important to show some statistical background as to how we obtained the results. We have followed the reviewer's suggestions and included error bars for 95% confidence intervals in Figure 1g (shown below). An edit has been made to Figure 1 and its caption in the main text as follows. We have also added an additional portion of discussion in the methods and Supplemental Information addressing the statistical analysis and replicate experiments.

Line 137: **(g)** The overall inactivation percentage over the x -axis and electric field strength as translated from the COMSOL Multiphysics simulation. In all our results presented, the entire region is analyzed and averaged with the duplicate values on the second half of the x -axis. The dashed blue lines are presented to guide the reader through a typical analysis process. The error bars indicate 95% confidence intervals.

Line 409: MATLAB was used to process the images and calculate the inactivation efficiency. The results from all viable channels out of a total 30 repeating units on one chip were averaged and used collectively to determine the inactivation percentage in all cases. With the symmetry of the

channels, there are a total of 120 data points (60 values, each with a duplicate) of electric field strengths resulted from each individual channel. Further details and MATLAB script are presented in Supplementary Text 2.5.

SI Line 170: This analysis results in 60 values for each electric field strength data point shown in the results. The bulk of the results in the text reflect single data points that were calculated as the average of all 60 values gathered from each of 30 replicate channels. The data analysis was developed specifically for consistency and simplicity of results presented to the readers.

2. Fig. 2: Again, error bars and number of replicates?

We appreciate the reviewer's feedback on the need for error bars and replicate experiment clarification. It is very important to us that the readers understand how our device was designed with replicates in mind and how we obtained our final results presented. Similar explanations are discussed in response 1 above (please refer to response 1). We originally chose not to show the error as Figures 2a-c are quite dense and including the error could be confusing to the readers. But in the best interest of the research, we have now calculated and included further information and error bars in the Supplemental Information. Each individual pulse width condition and Cu concentration depicted in Figures 2a-c have a figure showing the trend with error bars representing 95% confidence intervals for all analyzed replicates.

In traditional biological or chemical experiments there may be millions or even billions of cells/molecules analyzed, where only a few replicate experiments are decent enough for statistical analysis. For these specific lab-on-a-chip experiments, there are a limited number of cells (e.g., about 50-100 cells for each of the 120 columns) that we can analyze. Because of this limitation, we emphasize the importance of having a significant number of replicate channels like we do (30 per experiment). We also value the importance of statistical analysis and thank the reviewer again for pointing out our lack thereof in this work.

To address the reviewer's comments and offer a clearer understanding to the readers, we have included figures for error data in the Supplemental Information. In the updated Supplemental Information, we have shown the same curves as those shown in Figures 2a-c but in individual panels. There is no statistical analysis provided for Figures 2d-i, because the results shown are not pulled from the total replicate data but analyzed from the trends and relationships realized from the data points shown in Figures 2a-c. We have also included an edit to the Figure 2 caption to further clarify the rest of the figures and why statistical analysis is not provided more clearly to the readers. The revisions to the manuscript and SI are as follows and highlighted in the updated documents as well.

Line 172: The error bars indicating 95% confidence intervals for inactivation efficiency in Figures 2a-c are shown in in **Figs. S2-4** in the SI.

Line 178: There is no statistical analysis provided for Figures 2d-i as they are data analyzed and obtained from the data points, trends, and relationships shown in Figures 2a-c and not pulled from the total replicate data.

SI Line 53: **Figure S2 | Inactivation percentage for pulse width 500 ns and Cu concentrations 0-2 mg/L.** Error bars represent 95% confidence intervals for all analyzed replicates for each individual experiment for Cu concentrations (a) 0 mg/L, (b) 0.5 mg/L, (c) 1.0 mg/L, (d) 1.5 mg/L, (e) 2.0 mg/L.

SI Line 58: **Figure S3 | Inactivation percentage for pulse width 1 ms and Cu concentrations 0-2 mg/L.** Error bars represent 95% confidence intervals for all analyzed replicates for each individual experiment for Cu concentrations (a) 0 mg/L, (b) 0.5 mg/L, (c) 1.0 mg/L, (d) 1.5 mg/L, (e) 2.0 mg/L.

SI Line 63: **Figure S4 | Inactivation percentage for pulse width 2 ms and Cu concentrations 0-2 mg/L.** Error bars represent 95% confidence intervals for all analyzed replicates for each individual experiment for Cu concentrations (a) 0 mg/L, (b) 0.5 mg/L, (c) 1.0 mg/L, (d) 1.5 mg/L, (e) 2.0 mg/L.

3. It is very unusual in bioelectrics experiments to see such fine granularity in results as shown in Fig. 2 – just changing the electric field by less than 5% leads to vastly different inactivation efficacies. Usually, statistical differences obscure this sort of fine detail. Please discuss.

We appreciate the reviewer's comments on the lack of clarity on the results and statistical analysis. It is very important to us that the readers understand how our device was designed with replicates in mind and how we obtained our final results presented. The bulk of this and similar explanations are discussed in response 1 above (please refer to response 1). We have also updated the manuscript and Supplemental Information with further details regarding this comment and Figure 2 (please refer to response 2).

In addition to these explanations, we do understand the fine granularity of our data pointed out by the reviewer as unusual. Although it may be uncommon in bioelectric research, we have observed consistently that when the electric field strength reaches a certain threshold value, the inactivation efficiency observed can be very sensitive to small changes of electric field strength. We discuss this concept of the threshold value in our results but apologize for not making it clearer to the readers. We acknowledge that these trends do come with decent error bars (discussed in previous responses 1 and 2), but the trend is still obvious. With a small change over the threshold electric field strength, we do expect to see vast changes in the inactivation efficiencies. In all our experiments with varying parameters and conditions. We also clearly observe a strong trend every time and are confident in the fine and granular results present.

To further address the reviewer's concern, we have added an additional portion of explanation in the results, discussing the sensitive changes to inactivation efficiency after a threshold of electric field strength in each condition is crossed.

Line 150: This indicates that under these conditions, there is a threshold of electric field strength necessary to start electroporating bacteria regardless of pulse width. After this threshold is crossed, the inactivation values increase rapidly as shown in Figure 2a-c, where the inactivation efficiency observed is much more sensitive to changes in the electric field strength. Our detailed analysis results in consistent trends in all three conditions.

4. Fig. 3 – statistical analysis?

We appreciate the reviewer's comments on the lack of clarity on the results and statistical analysis. It is very important to us that the readers understand how our device was designed with replicates in mind and how we obtained our final results presented. This and similar explanations are discussed in response 1 and 2 above (please refer to response 1 and 2).

Regarding Figure 3 specifically, we apologize for not being clearer to the readers. Figure 3 is composed of results for three different time series experiments, so it varies compared to the experiments/results conducted/presented in previous portions of the paper. For these results, the same device is used, but due to time and processing constraints, there are only 5 replicate channels for each experiment and therefore 10 inactivation values collected for each electric field strength measured. The images of one representative channel for each type of experiment is shown in Figure 3a-c. For the data presented in Figure 3d, the individual data points are calculated from the average of 5 different channels and therefore 10 different replicate points. We have taken the reviewer's suggestion and calculated the error for these data and updated Figure 3d and the supplemental plots accordingly. The error bars indicate 95% confidence intervals, but due to the very limited number of cells analyzed and the added limitation of only being able to have 5 replicate channels (compared to the usual 30) we acknowledge the larger error bars here. Although the error is relatively large due to the fewer replicate channels and data obtained, we believe the trend is still very clear and obvious to the readers.

To further address the reviewer's comments, we have edited a portion the caption of Figure 3 addressing the replicate experiments, error bars, and limitations for the data and images shown. The revisions to the manuscript and SI are as follows and highlighted in the final document as well.

Line 240: Images for 1 minute into the experiment are also presented for each condition to show inactivation immediately following any pulse application. Due to time and processing constraints, there are only 5 replicate channels for each experiment collected and analyzed in the time series results presented (compared to the usual capacity of 30). Only one representative channel for each experiment is shown. The zoomed in portions are shown for 1 and 180 minutes for each condition,

and areas of significance are highlighted by yellow dashed circles to guide the readers. **(d)** The overall inactivation percentage over time in minutes for Cu-only, EFT-only, and EFT-Cu. The dashed line refers to the calculated theoretical additive. Individual plots are shown for specific electric field strengths appearing in increasing order from left to right and top down. The error bars represent 95% confidence intervals for these and all additional plots with electric field strengths 29-39 kV/cm. (For any references to colors in the figures the readers are referred to the online/web version of this article).

SI Line 68: **Figure S5 | Additional and all plots for quantified inactivation percentage taken every 30 minutes over the course of 3 hours between 30-38 kV/cm electric field strength.** The overall inactivation percentage over time in minutes for Cu-only, EFT-only, and EFT-Cu where the theoretical additive is calculated and compared to the measured EFT-Cu result. Individual plots are shown for specific electric field strengths, appearing in increasing order from top down and left to right. The error bars represent 95% confidence intervals. (For any references to colors in the figures the reader is referred to the online/web version of this article).

5. Another mechanism to consider (or at least mention) is electrophoresis. See this paper for an example: Li J, Lin H. Numerical simulation of molecular uptake via electroporation. *Bioelectrochemistry*. 2011 Aug 1;82(1):10-21.

We want to thank the reviewer for the paper suggestion and additional mechanism to consider. After carefully reading through the article provided by the reviewer, we believe there is interesting discussion from it that can further our understanding and scope of the research accomplished as we use charged ions in our experiments and standard PI dye methods for observing cell staining and death with electroporation. The paper discusses how electrophoresis and diffusion can both play a role in transport and uptake of small molecules into the cell. Although it is interesting to consider the mechanism of Cu ion uptake by cells through electrophoresis, after careful consideration, we believe this is not a significant mechanism for our study. In using charged calcium ions and CaFluo dye, Li et al. concludes the electrophoresis effects are significant for time less than 6 ms, and anything following is by diffusion.¹ As electrophoresis can be considered for when pulses are being applied, there may be maximum 20 seconds initially in our experiments (length of time for pulses to finish applying) where the mechanism could be considered, but there are 2-3 hours following where diffusion would remain the main mechanism for Cu or PI uptake. Because our experiments not only focus on the short period of time EFT is applied, but the effects of Cu ions for much longer times with no pulses applied, we believe electrophoresis is not a significant mechanism to consider in this study. Despite this, the paper confirms that the diffusion of PI into cells continues after pulses ceased and the enhancement was stronger in this second half observed. It also emphasizes that the diffusion of PI can vary inversely with conductivity of the solution. We have made great efforts to maintain all our varying Cu concentration solutions at the same conductivity for this purpose.

Following the reviewer's suggestion, we have added an additional portion of text to the methods to include the discussion and findings relevant from the article the reviewer has recommended. The revisions to the manuscript are as follows and highlighted in the final document as well.

Line 300: After pulses are removed in EFT-only, we observe PI staining to continue for a very short period (few seconds). This is concurrent with literature as Li et al. confirmed diffusion of PI into cells continue after pulses are ceased with fluorescence intensity increasing shortly after pulses were removed.¹

Line 386: As the conductivity of solution can alter the strength of electric field applied and the uptake of PI into the cell, all Cu solutions were made with the addition of sodium sulfate to ensure the conductivity of all solutions resulted were the same as that of the 2 mg/L solution (approximately ~10 mS/cm) for consistency.

Line 404: For all cases, the cell viability was indicated using standard, well-established propidium iodide (PI, 2 mM) staining. The PI staining method is used as PI molecules will only enter dead bacteria with a compromised membrane and bind to its DNA to result a fluorescence enhanced several folds as reiterated by many other studies focusing on electroporation and cell death.^{1,2}

6. Consider citing this manuscript outlining synergistic treatments of microorganisms with pulsed electric fields as it seems particularly relevant to the last paragraph of the manuscript: Garner AL. Pulsed electric field inactivation of microorganisms: from fundamental biophysics to synergistic treatments. *Applied microbiology and biotechnology*. 2019 Oct;103(19):7917-29.

We want to thank the reviewer for the paper suggestion. After taking a look at the article provided, we believe it offers many additional studies that are of interest for perspective on our future research. It also increases our knowledge of how we can go about synergistic studies as they may be relevant to other works and potential future applications. Great promise has been shown for the combination of both physical and chemical-based methods using pulsed electric fields in food research groups, and we believe these similar effects could be amplified in future research with EFT and Cu.²

Following the reviewer's suggestion, we have added additional portions of discussion in the last paragraph of the manuscript with citing the paper the reviewer provided. The revisions to the manuscript are as follows and highlighted in the updated document as well.

Line 347: Future work can test other conditions and parameters as reviewed by Garner AL in more traditional pulsed electric field treatments, *e.g.*, a wider range of pulse widths, different total effective treatment times, and effects of conductivity, pH, temperature, and turbidity of the liquid.² Finally, great importance should be placed on studying the potential enhanced and synergistic effects of combining EFT with other established disinfectants and methods of disinfection as great promise is shown for the combination of both physical and chemical based methods using EFT-Cu.

7. With 40,000 pulses applied, how was sample heating controlled? Heating can have a significant effect on inactivation, and it is important to at least qualitatively understand the relative contribution of heating and electric field effects.

We appreciate the reviewer's comment and concern for heating within the experimental setup. We agree with the reviewer that heating, along with ROS and bubble formation are all important factors to be avoided in order to independently focus on the variables tested. Along with this, we believe a small amount of heating, similar to ROS, is unavoidable in pulses electric field experiments. We understand the concern for heating in our studies as it can significantly impact the results of the inactivation efficiency as alluded in previous papers by Garner and Song et al.^{2,6} These authors state applying pulses can increase the temperature of the surrounding solution and that applying another pulse before there is proper time for thermal dissipation will then increase the temperature of the solution for the next pulse. We agree this may be of concern but have taken proper precautions to limit this as much as possible in our experiments. Garner et al. has also done specific experiments with nano- and microsecond pulses, recording the temperature gradient across the cell membrane and the time to dissipate this thermal effect.⁷ They found that the temperature shift induced by pulse durations of 10 μ s dissipates over time and that at the longest time of 85 μ s they tested, the temperature difference becomes ~ 0 (no significant heating effects). These are pulse widths with rest periods of less than 1 order of magnitude, resulting in no significant temperature gradients. In all our experiments, the operating parameters are set to have a duty cycle of 1:1000. This means our rest time for any heat impacts to dissipate between pulses are 3 orders of magnitude longer than the individual pulse applied for all conditions.

In previous studies using similar devices and conditions, there were additional finite element simulations conducted to evaluate the temperature of the systems and found that heating is not likely to induce any cell inactivation. These previous studies also concluded the temperature increase, if any, to be marginal and dissipate with the elongated periods in between pulses.³ It is because of these previous works that we have designed our experimental parameters with duty cycles of 1:1000 in mind. These parameters were also determined to specifically limit and avoid factors like bubble formation and ROS generation, along with any significant heating as they can all impact the results. Our devices are also sealed and encased with a water-based solution during pulse application in order to ensure the cells do not dry out. Not only is the water difficult to heat significantly, but we do not observe any signs of heating (no bubble formation, rapid evaporation of surrounding liquid, chip is still cold to the touch).

We thank the reviewer for pointing out heating was not explicitly mentioned in our study. To be clear, with our preventative measures in place, we are confident there are no concerns on our end for significant heating on the device. While we agree, significant heating can have sizable effects on inactivation, we observe no heating during our operando experiments and are further reassured that even with 40,000 pulses applied, we can still achieve a wide range of cell death either 0%, minimal, or high inactivation, without any significant observation of heating, ROS generation, or bubble formation.

To address the reviewer's concern, we have added an additional portion of discussion in the methods to explicitly explain why we work to minimize heating effects and believe heating is not a significant concern in this study. The revisions to the manuscript are as follows and highlighted in the final document as well.

Line 396: The pulses were applied to the LOAC device using a high-speed pulse generator (Avtech AV-1010-B) triggered with a waveform generator (Keysight 33509B) (further details on parameter setup, procedure of pulse applications, and avoiding heating effects, bubble, and ROS formation are discussed in Supplementary Text 2.4).

SI Line 140: For all experiments conducted in this study, it was important to avoid reactive oxygen species (ROS) and bubble generation, along with any significant heating effects during the application of electric pulses.

SI Line 153: In addition, all our experiments used operating parameters set to have a duty cycle of 1:1000. This means our rest time for any heat impacts to dissipate between pulses are 3 orders of magnitude longer than the individual pulse applied for all conditions. These parameters were determined through previous and preliminary studies to specifically limit and avoid factors like bubble formation and ROS generation, along with any significant heating as they can all impact the results. With these preventative measures in place, we are confident there is no concerns for heating on the device. All EFT-Cu experiments were performed to limit ROS generation, bubble formation, and heating so that we can be confident in the understanding of how Cu and EFT can work both independently and together without the interference of any other inactivation mechanisms potentially present.

REFERENCES

- 1 Li, J. & Lin, H. Numerical simulation of molecular uptake via electroporation. *Bioelectrochemistry* **82**, 10-21 (2011).
[https://doi.org:https://doi.org/10.1016/j.bioelechem.2011.04.006](https://doi.org/https://doi.org/10.1016/j.bioelechem.2011.04.006)
- 2 Garner, A. L. Pulsed electric field inactivation of microorganisms: from fundamental biophysics to synergistic treatments. *Applied Microbiology and Biotechnology* **103**, 7917-7929 (2019). <https://doi.org:10.1007/s00253-019-10067-y>
- 3 Wang, T., Brown, D. K. & Xie, X. Operando investigation of locally enhanced electric field treatment (LEEFT) harnessing lightning-rod effect for rapid bacteria inactivation. *Nano Letters* **22**, 860-867 (2021).
- 4 Wang, T., Chen, H., Yu, C. & Xie, X. Rapid determination of the electroporation threshold for bacteria inactivation using a lab-on-a-chip platform. *Environment international* **132**, 105040 (2019).
- 5 Wang, T. & Xie, X. Nanosecond bacteria inactivation realized by locally enhanced electric field treatment. *Nature Water* **1**, 104-112 (2023). <https://doi.org:10.1038/s44221-022-00003-2>
- 6 Song, J., Joshi, R. P. & Schoenbach, K. H. Synergistic effects of local temperature enhancements on cellular responses in the context of high-intensity, ultrashort electric pulses. *Medical & Biological Engineering & Computing* **49**, 713-718 (2011).
<https://doi.org:10.1007/s11517-011-0745-z>
- 7 Garner, A. L. *et al.* Cell membrane thermal gradients induced by electromagnetic fields. *Journal of Applied Physics* **113** (2013). <https://doi.org:10.1063/1.4809642>

REVIEWER COMMENTS

Reviewer #1 (Remarks to the Author):

Thank you authors for the nice and detailed answer. The manuscript was improved but I would stay with the main concern about the viability. Membrane permeability is not the same as viability. Moreover, the synergetic effect of electroporation was explored quite well by using less harmful materials, like here <https://doi.org/10.1186/s12866-019-1447-1>. Even after a lot of revision work, I am not convinced that this work is worthy of publication in this journal.

Reviewer #2 (Remarks to the Author):

The authors have addressed my prior comments but, in doing so, introduced a more difficult to address issue (which was commented on by Reviewer #1 initially). Using PI as a viability stain is inherently problematic when you stress the membrane [cf., Davey HM, Hexley P. Red but not dead? Membranes of stressed *Saccharomyces cerevisiae* are permeable to propidium iodide. *Environmental microbiology*. 2011 Jan;13(1):163-71.]. This must be addressed further prior to publication, with data if possible. More details follow:

1. Thank you for the clarification on how the chips were made. I am impressed about being able to have essentially 30 replicates on the chip. This would be very helpful to emphasize in the figure captions [“The error bars indicate 95% confidence intervals based on 30 replicates from a single chip” or something like that].
2. Thank you also for the insight on publication venue. The multidisciplinary aspect seems like a relevant reason for this journal choice, particularly given the novelty of the method under investigation.
3. Thank you for clarifying the temperature effects – this will be important in practical application.
4. I disagree with the statement of PI only entering a dead microorganism. This is almost certainly false – the part about entering through a compromised membrane is valid, but a compromised membrane does not necessarily imply cell death. Applying pulsed electric fields almost certainly permeabilizes the

membrane based on my understanding of the mechanism involved. Refs. [46] and [47] do not state this explicitly, as the author suggest in Line 407. The review paper by Garner only addresses PI as a membrane integrity dye, not for viability. Li and Lin also make no specific mention of PI being used for viability staining, although the authors application of their modeling to assess electrophoresis and diffusion is correct. One could argue that membrane permeabilization leads to greater copper delivery to the cells, causing cell death; however, I think stating that this definitively shows cell death is problematic.

The use of dyes for assessing membrane integrity has been a challenge in bioelectrics for decades. For instance, early works using nanosecond pulses suggested that the membrane was not permeabilized because ethidium homodimer, a common membrane integrity dye, did not penetrate. Later experiments using a smaller dye (YO-PRO-1) indicated that smaller pores were formed by these nanosecond pulses.

Periodically, experiments with mammalian cells showed that trypan blue uptake, which is commonly associated with cell death, could occur following intense electric pulsed due to membrane permeabilization (this was only determined because uptake at later times was lower due to pore resealing).

The point of my writing this is to point out that (and, to a degree, sympathize with) assessing viability with a membrane exclusion dye is difficult. In conventional electroporation (mammalian cell) experiments, PI is often used for membrane integrity and trypan blue for viability, but, as I pointed out above, this must be assessed carefully. In fact, dead cells may be impermeant to PI (<https://www.ncbi.nlm.nih.gov/pmc/articles/PMC4178667/>). Here is another review on assessing microorganism health that may be helpful: <https://doi.org/10.3389/fmicb.2020.547458>.

There are viability tests that leverage PI, such as this one: Stiefel, P., Schmidt-Emrich, S., Maniura-Weber, K. et al. Critical aspects of using bacterial cell viability assays with the fluorophores SYTO9 and propidium iodide. *BMC Microbiol* 15, 36 (2015). <https://doi.org/10.1186/s12866-015-0376-x>. But again, PI cannot be used alone when the membrane is stressed: Davey HM, Hexley P. Red but not dead? Membranes of stressed *Saccharomyces cerevisiae* are permeable to propidium iodide. *Environmental microbiology*. 2011 Jan;13(1):163-71.

The best way (or, maybe, a more reasonable way) would be some sort of molecular test, but this seems not to be possible here. The authors stated the idea of looking at PI a long time after treatment. That may help to mitigate the membrane permeabilization issue to an extent since many pores will have sealed. However, this is a significant issue.

Operando Investigation of the Synergistic Effect of Electric Field Treatment and Copper for Bacteria Inactivation

Mourin Jarin¹, Ting Wang¹, Xing Xie^{1,2*}

¹School of Civil and Environmental Engineering, Georgia Institute of Technology, Atlanta, Georgia 30332, United States

²Institute for Electronics and Nanotechnology, Georgia Institute of Technology, Atlanta, Georgia, 30332, United States

*Corresponding author: Xing Xie, Email: xing.xie@ce.gatech.edu, Phone: (404)894-9723

We want to thank the reviewers again for the constructive and helpful comments. Their insights and feedback have allowed us to improve the manuscript. Below, we have addressed the main concern of cell viability pointed out by both reviewers in detail.

Comments from Reviewers

Reviewer #1

Thank you authors for the nice and detailed answer. The manuscript was improved but I would stay with the main concern about the viability. Membrane permeability is not the same as viability. Moreover, the synergetic effect of electroporation was explored quite well by using less harmful materials, like here <https://doi.org/10.1186/s12866-019-1447-1>. Even after a lot of revision work, I am not convinced that this work is worthy of publication in this journal.

Reviewer #2

The authors have addressed my prior comments but, in doing so, introduced a more difficult to address issue (which was commented on by Reviewer #1 initially). Using PI as a viability stain is inherently problematic when you stress the membrane [cf., Davey HM, Hexley P. Red but not dead? Membranes of stressed *Saccharomyces cerevisiae* are permeable to propidium iodide. *Environmental microbiology*. 2011 Jan;13(1):163-71.]. This must be addressed further prior to publication, with data if possible. More details follow:

1. Thank you for the clarification on how the chips were made. I am impressed about being able to have essentially 30 replicates on the chip. This would be very helpful to emphasize in the figure captions [“The error bars indicate 95% confidence intervals based on 30 replicates from a single chip” or something like that].
2. Thank you also for the insight on publication venue. The multidisciplinary aspect seems like a relevant reason for this journal choice, particularly given the novelty of the method under investigation.
3. Thank you for clarifying the temperature effects – this will be important in practical application.
4. I disagree with the statement of PI only entering a dead microorganism. This is almost certainly false – the part about entering through a compromised membrane is valid, but a compromised membrane does not necessarily imply cell death. Applying pulsed electric fields almost certainly permeabilizes the membrane based on my understanding of the mechanism involved. Refs. [46] and [47] do not state this explicitly, as the author suggest in Line 407. The review paper by Garner only addresses PI as a membrane integrity dye, not for viability. Li and Lin also make no specific mention of PI being used for viability staining, although the authors application of their modeling to assess electrophoresis and diffusion is correct. One could argue that membrane permeabilization leads to greater copper delivery to the cells, causing cell death; however, I think stating that this definitively shows cell death is problematic.

The use of dyes for assessing membrane integrity has been a challenge in bioelectrics for decades. For instance, early works using nanosecond pulses suggested that the membrane was not permeabilized because ethidium homodimer, a common membrane integrity dye, did not penetrate. Later experiments using a smaller dye (YO-PRO-1) indicated that smaller pores were formed by these nanosecond pulses.

Periodically, experiments with mammalian cells showed that trypan blue uptake, which is commonly associated with cell death, could occur following intense electric pulsed due to membrane permeabilization (this was only determined because uptake at later times was lower due to pore resealing).

The point of my writing this is to point out that (and, to a degree, sympathize with) assessing viability with a membrane exclusion dye is difficult. In conventional electroporation (mammalian cell) experiments, PI is often used for membrane integrity and trypan blue for viability, but, as I pointed out above, this must be assessed carefully. In fact, dead cells may be impermeant to PI (<https://www.ncbi.nlm.nih.gov/pmc/articles/PMC4178667/>). Here is another review on assessing microorganism health that may be helpful: <https://doi.org/10.3389/fmicb.2020.547458>.

There are viability tests that leverage PI, such as this one: Stiefel, P., Schmidt-Emrich, S., Maniura-Weber, K. et al. Critical aspects of using bacterial cell viability assays with the fluorophores SYTO9 and propidium iodide. *BMC Microbiol* 15, 36 (2015). <https://doi.org/10.1186/s12866-015-0376-x>. But again, PI cannot be used alone when the membrane is stressed: Davey HM, Hexley P. Red but not dead? Membranes of stressed *Saccharomyces cerevisiae* are permeable to propidium iodide. *Environmental microbiology*. 2011 Jan;13(1):163-71.

The best way (or, maybe, a more reasonable way) would be some sort of molecular test, but this seems not to be possible here. The authors stated the idea of looking at PI a long time after treatment. That may help to mitigate the membrane permeabilization issue to an extent since many pores will have sealed. However, this is a significant issue.

Response to Reviewers' Comments:

We appreciate the reviewers' overall assessment and recognition of our research and findings in the first round of review. We also appreciate the reviewers' consensus that we have addressed most of their concerns and improved the manuscript. We additionally want to thank the reviewers for pointing out one more critical area of concern that we can still improve upon. The following presents a detailed breakdown of our response to the main concern regarding cell viability and also includes our adjustments to the manuscript.

We agree with the reviewers that the general concept of "red" equals "dead" regarding PI staining is false. We apologize for the incorrect information we may have suggested in the manuscript and previous response. We have also gone ahead and adjusted this text in the main manuscript accordingly. We acknowledge that both reviewers make excellent points that the topic of cell viability was not addressed well originally in the manuscript or explained clearly to the readers.

Reviewer 1 provided a paper by Novickji *et al.* that clearly identifies the PI staining of the same model bacteria we use *S. aureus* resulting in different results for permeabilization versus viability.¹ The authors conducted their pulses experiments using a square wave electroporator and commercially available cuvettes. Although having very minimal volume, they could obtain enough solution to culture the cells and calculate cell viability this way. Our lab is similarly using a cuvette system to investigate the combination of electric field treatment (EFT) and copper (Cu) disinfection, where both plate count and molecular tests will be accomplished in the analysis. However, the study is still at the early stage, and we will publish the results later separately. Using the lab-on-a-chip (LOAC) device in this work, unfortunately, we are limited in our ability to do the same cell viability analysis. Nevertheless, we agree this paper by Novickji *et al.* provides very important results we have considered with the current revision of our main text.

Reviewer 2 provided several papers also very helpful to our understanding of this issue. The first by Davey *et al.* focusing on how cell membranes of *S. cerevisiae* are permeable to PI during/after various forms of stress are induced, but not necessarily inactivated.² This paper reported 7% of cells tested could be cultured after exhibiting temporary PI permeability irrespective of the stress condition. The authors importantly indicated that we must be carefully considerate when using PI-positively synonymously with dead cells, which we agree we did not do carefully enough initially. The authors then went on to state that it would be necessary to remove the stress (pulses in this case) and allow for a period of recovery (*i.e.*, rest time) prior to addition of the stain. We agree with this paper that adding stain before/during the experiments can result in inaccurate data and findings, specifically for EFT experiments, where reversible electroporation needs to be addressed. Nevertheless, we want to clarify with the reviewers that we did consider this important point in our methodology originally as explained below.

Before beginning this study, we conducted many preliminary experiments to determine the best operating parameters and methodology. As previously mentioned, we tried to mitigate factors such as reactive-oxygen-species (ROS) generation, bubble formation, and also heating effects kindly pointed out by Reviewer 2. These preliminary studies were also used to determine whether we would need the use of a double staining method (described below). As for some of these types of studies, a double staining method can aid in limiting the uncertainty for reversible electroporation and inaccuracy of results. Previous studies using similar LOAC devices for mechanism studies of EFT by our lab have used both single staining with PI and double staining methods depending on the experimental procedure.^{3,4}

For the double staining method we applied before, SYTOX Green staining was first used to show any damaged cells from the pulsed electric fields. PI staining was then used after 20 minutes to assume any reversible pores would have already closed, leaving the cells stained red/orange/yellow to be permanently damaged and inactivated.^{3,5} Our preliminary tests for this study involved a similar double staining method, but a limitation we found with having a wider view of fluorescence to analyze was the potential inaccuracies that came with having more than one dye present at a time. Reviewer 2 also pointed this out with a referenced paper by Stiefel *et al.* that discussed this similar issue that with a double staining method (using the same 2 dyes SYTOX Green and PI), the mixture could alter the intensities of the individual dyes and add background values leading to inaccurate measurements of PI signals in the end.⁶

Another study using the single staining method performed parallel experiments to assess the significance, if any, of reversible electroporation in their results. It was found that adding PI stain before pulses were applied or 2 hours after resulted in no significant difference in the efficiency of the measured inactivation values (indicated by the positive staining) in the end.⁴ Therefore, we decided a double staining method was not necessary for this work as long as the dye was applied after long enough rest period. For all the experiments conducted to obtain the results shown in Figure 2 of the manuscript (the bulk of our research), the cells on the LOAC device were initially treated, then rested for 2 hours to allow reversible-pore closure and/or Cu ion inactivation,^{5,7} and lastly stained with PI before imaging at the end of the process. With this method, we are confident that the positive PI staining is a reliable indication of permanent cell damage, *i.e.*, inactivation in this study. Although it was mentioned very shortly in the previous versions of the manuscript, we apologize that it was not discussed more in depth initially. Now, we have made it much clearer in the revised manuscript.

For Figure 3, the purpose was to observe a time series of results, meaning PI stain was added at the beginning of the experiment before pulses were applied. This means we cannot guarantee the same validity of inactivation in initial treatment portion. We want to emphasize that the EFT in our study, if any is present, only occurred within the first 20 seconds of the experiment. We agree that in these experiments, some of the cells might be stained due to reversible electroporation and might not have been heavily damaged or totally inactivated. Although this was a concern of ours initially, we would like to point out that for the EFT-only condition in Figure 3b (shown below as Figure R1), there is no significant increase in fluorescent cells over the course of the total 3 hours after the first 1-minute image, taken after pulses were removed. This trend was also observed when calculating the results as shown in Figure 3d (shown below as Figure R2). Looking specifically at the EFT-only condition (grey line), it shows a consistently linear and flat line trend for every electric field strength and time analyzed. These results clearly indicate that PI staining caused by any reversible electroporation (*i.e.*, false inactivation) can only occur during or very shortly after pulses are applied. This again supports our logic that a 2-hour rest period is enough time for reversible pore closer, as already backed by the previous studies and literature referenced. In addition, the inactivation percentages at the time stamp of 2 hours shown in Figure 3b (see below as Figure R3), where PI was added before pulses were applied, are very similar to those shown in Figure 2a (see below as Figure R4) under the same treatment conditions but with PI added 2 hours later. This indicates further that the PI staining due to reversible electroporation is not significant as predicted by our preliminary tests, and therefore not of high concern for the bulk of our experimental data. Nevertheless, we understand the dye staining method used in results for Figure 3 is not consistent to our methods for the results obtained for Figure 2 and should be clarified further for the readers' understanding. Because of this, we have adjusted our discussion of the results to reflect this accordingly with a disclaimer of the change in methods and emphasis on membrane permeability rather than only inactivation in the first minute where pulses are applied.

Fig. R1 (Figure 3b from the main text) | Microscopy image of the LOAC device under fluorescent channel taken every 30 minutes over the course of 3 hours. EFT-only using EFT + 0 mg/L Cu is shown for reference to the reviewer. The pulse width for EFT is operated under 500 ns.

Fig. R2 (Figure 3d from the main text) | LOAC device quantified inactivation percentage over the course of 3 hours. The overall inactivation percentage over time in minutes for Cu-only, EFT-only, and EFT-Cu. The dashed line refers to the calculated theoretical additive. Individual plots are shown for specific electric field strengths appearing in increasing order from left to right and top down. The error bars represent 95% confidence intervals for these and all additional plots with electric field strengths 29-39 kV/cm.

Fig. R3 (for reviewers' reference only) | Inactivation percentage for pulse width 500 ns and Cu concentrations 0 mg/L at the 2-hour mark from the time series experiments (original results presented in Figure 3 of the main text). Error bars represent 95% confidence intervals for all analyzed replicates for each individual experiment.

Fig. R4 (Figure S2a from the SI) | Inactivation percentage for pulse width 500 ns and Cu concentrations 0 mg/L. Error bars represent 95% confidence intervals for all analyzed replicates for each individual experiment.

Regarding Figure 4 where the single cell analysis is observed operando, there is also PI stain present during the pulse application. This part of the study is only focused on the individual cell, its cell membrane permeability, and initial rate of the observed fluorescence. Because of this, we are not concerned with the reversible pore closure or inactivation but have also adjusted the discussion text regarding Figure 4 accordingly as we agree strongly with the reviewers' comments.

To summarize, the bulk of our data and results have already considered the cell viability issue and reversible pore closure and accountant for it best we could in the methodology. We are happy to make further changes to the manuscript and its original discussions as we agree with the reviewers that some of our initial statements were incorrect and not clarifies enough. With the main limitation of the chip being that we cannot account for cell viability in any other way, we believe the edits made in the text and clarifications made to the readers are the best method going forward. With all of this explained and adjusted for, we still firmly remain confident in our results and the methodology that they were conducted in. Again, we sincerely apologize for not making our methodology and reasoning for the single staining process clearer in the beginning as we did not believe this was necessary as it is complex and may potentially confuse the reader in the main text. We now understand this detail is very important to the understanding and contribution of our work and have adjusted the manuscript methods, results, and SI accordingly. We ultimately believe this study remains a strong and solid piece of work, with novelty, that can contribute important impact to the journal and multidisciplinary fields.

We want to end with stating that if the reviewers still have additional concerns, we are more than willing to adjust the language used in the manuscript further. Please feel free to let us know the language you may feel is more accurate or appropriate to use regarding the cell viability concerns of the work in a direct recommendation and we will be happy to implement it.

Line 116: **Fig. 1f** demonstrates the fluorescent red, inactivated cells across the electric field gradient along the channel. The inactivation here refers to positive PI-staining further explained in the methods section.

Line 138: (g) The overall inactivation percentage over the x-axis and electric field strength as translated from the COMSOL Multiphysics simulation. The inactivation here refers to positive PI-staining further explained in the methods section. In all our results presented, the entire region is analyzed and averaged with the duplicate values on the second half of the x-axis. The dashed blue lines are presented to guide the reader through a typical analysis process. The error bars indicate 95% confidence intervals based on 30 replicated from a single chip.

Line 172: **Fig. 2 | Microbial inactivation of combined electric field treatment and copper (EFT-Cu).** (a-c) The overall inactivation percentage over the electric field strength for five varied Cu concentrations 0-2 mg/L are presented when the pulse width is 500 ns (a), 1 μ s (b), and 2 μ s (c). The overall inactivation here refers to positive PI-staining further explained in the methods section. The error bars indicating 95% confidence intervals for inactivation efficiency in Figures 2a-c are shown in **Figs. S2-4** in the SI.

Line 207: The fluorescent images of the center portion of the device channels during these three treatment conditions over time are shown in **Figs. 3a-c**, respectively. Overall inactivation here refers to positive PI-staining further explained in Supplementary Text 2.5.

Line 243: Images for 1 minute into the experiment are also presented for each condition to show cell damage immediately following any pulse application. The overall inactivation here refers to positive PI-staining further explained in Supplementary Text 2.5.

Line 256: In this section, the focus was on observing the membrane permeability in a singular cell in each of the three conditions mentioned in Section 2.4 *operando*. To visually compare, **Figs. 4a-c** shows the images of a single cell in each of the three conditions. The cells either show fluorescence from Cu ion permeation alone (**Fig. 4a**), electroporation damage through EFT (**Fig. 4b**), or the combination effect where EFT is applied first, and damage to the cell occurs from Cu ion permeation after pulses are removed (**Fig. 4c**) (short videos for each condition are linked and described in Supplementary Videos). The quantified intensity of dye saturation for each image is also normalized over time as shown in **Figs. 4d-f** along with another individual cell showing similar results. **Fig. 4a** shows the dye staining for Cu-only, where time 0 represents the moment before any fluorescent intensity is observed or measured, and full saturation (intensity 100%) is measured and observed at \sim 44 s. **Fig. 4d** further confirms a slow and steady increase from Cu ion permeation for Cu-only. In **Fig. 4b** for EFT-only, the cell shown in the images is exposed to a high electric field strength region and quickly damaged through electroporation. According to our results from the previous sections, the EFT-only condition can achieve \sim 40% overall inactivation, while the other \sim 60% of cells continue to observe no damage. The single cell shown in **Fig. 4b** is specifically one of the \sim 40% of cells most likely inactivated through EFT alone. It is important to note that the cell is infiltrated with dye rapidly here and measures complete saturation at only \sim 3.3 s as indicated by the non-linear trend in **Fig. 4e**. In **Fig. 4c** for EFT-Cu, the cell is exposed to EFT while dosed with Cu ions, but not observed to be fully saturated with dye until several minutes after pulses are removed. Different to the EFT-only case, some of the other \sim 60% of cells still intact after pulses are removed, are observed to be damaged here later on by the Cu ions in solution using the combination approach of EFT and Cu. The image frames for EFT-Cu show the cell is saturated with dye at a faster pace than for Cu-only as the cell measures and observes complete

saturation within ~21 s, or less than half the time. EFT-Cu shows a measured trend (Fig. 4f) comparative to the combination of EFT-only and Cu-only, a slow and steady start with a sharp increase to reach complete saturation. This result suggests there is an observable difference in the rate the cell membranes are being damaged using these two methods independently versus when they are combined.

Line 283: **Fig. 4 | Microscopy images of the LOAC device under fluorescent and DIC channels where the red indicates PI fluorescence and diffusion of the dye into the individual cells and the relative time points where each image's time stamp is shown underneath. (a-c)** Cu-only using no EFT + 2 mg/L Cu (a), EFT-only using EFT + 0 mg/L Cu (b), and EFT-Cu where EFT + 2 mg/L Cu (c) are all shown. The pulse width for EFT is operated under 2 μ s. **(d-f)** The relative PI intensity of fluorescence as measured by the microscope normalized over time in seconds is also shown for Cu-only (d), EFT-only (e), and EFT-Cu (f) where importance is placed on when the cells reach 100% saturation of the dye intensity. (For any references to colors in the figures the reader is referred to the online/web version of this article).

Line 293: From our understanding of EFT and Cu's independent mechanisms to inactivate bacteria and the observed results of the single cell study, there is further evidence pointing to an increased cell permeability as a result of the applied EFT. For Cu-only (Fig. 4a), the cells show successful membrane damage with PI staining, but the process to achieve complete staining takes long (~44 s). This is concurrent with literature on copper induced contact killing of microorganisms, as it may take minutes to hours for inactivation to occur (Fig. 3a for Cu only inactivation ranges from 1 minute to 3 hours).^{8,9} Studies have correlated this cell damage and death with lipid peroxidation, loss of membrane integrity, and Cu ion uptake.¹⁰⁻¹² These studies also agree that DNA degradation is not the primary cause of cell death by Cu, but rather the gradual release of genetic material from membrane damage.^{11,12} For EFT-only (Fig. 4b), this process occurs in under 10 s (~3.3 s) for higher electric field strength regions. When the electric field strength is strong enough and applied for long enough, it will cause damage to the cell membrane or create pores much faster, allowing for rapid diffusion of the dye and quicker inactivation as shown.¹³ After pulses are removed in EFT-only, we observe PI staining to continue for a very short period (few seconds). This is concurrent with literature as Li et al. confirmed diffusion of PI into cells continue after pulses are ceased with fluorescence intensity increasing shortly after pulses were removed.¹⁴ When the combination approach EFT-Cu is applied, there are some cells that stain rapidly from the EFT cell damage, and numerous others that continue to stain over the next few minutes to hours. After the pulses are applied, these cells are achieving complete staining at a rate faster than that of only Cu ion disinfection (Fig. 4c), indicating the application of the EFT is damaging or weakening the membrane for cells that have not yet been damaged from electroporation alone. Since exposing bacteria to external electric fields can alter their structural integrity, this result suggests EFT is increasing the permeability of the cell membrane for Cu ions to act more quickly in creating more permanent damage to the cells and inevitably cause cell death.¹⁵⁻¹⁷ Not only is the diffusion of the dye faster, but also the number of bacteria that continue to fluoresce over time increases (confirmed in earlier sections). We are confident the combined approach has an enhancement effect leading bacteria to be weakened by the EFT and more susceptible to inactivation by Cu ion permeation. To our knowledge, this exciting observation is the first-time researchers are able to visually conclude the synergistic effects of EFT-Cu at microscale.

Line 408: For the bulk of the experiments, after Cu solution and/or electric field was applied, there was a 2-hour effective treatment time and rest period after pulses were removed to best collect the

most representative results at this scale. From our preliminary studies, we found that there was no significant difference in the results obtained using a single staining or double staining method and a single staining method would also result fewer potential inaccuracies, and thus, a single staining method was used. Most of the data analysis was done through quantifying the number/percentage of cells inactivated, over time, and in specific electric field regions. For all cases, the results were indicated using standard, well-established propidium iodide (PI, 2 μ M) staining. PI molecules will only enter bacteria with a compromised membrane and bind to their DNA to result a fluorescence enhanced several folds as reiterated by many other studies on electroporation and membrane permeability.^{14,18,19} In case any reversible electroporation was present from the pulse application, the cells were only stained at the end of the experiment, after treatment, the 2 hour rest period, and right before imaging. Since any reversible membrane damages should be recovered after the 2-hour period, we consider the cells stained with PI as inactivated for the bulk of the results presented (Figure 2).^{5,7} The results presented in Figure 3&4 regarding the time series and single cell analysis have a slightly different methodology and are further detailed in the Supplementary Text 2.5.

SI Line 37: 2.5 Imaging and data processing through MATLAB

SI Line 39: Video S1. PI staining of cells treated with Cu-only

SI Line 40: Video S2. PI staining of cells treated with EFT-only

SI Line 41: Video S3. PI staining of cells treated with EFT-Cu after pulses are removed

SI Line 42: Video S4. Single cell analysis through PI staining with Cu-only

SI Line 43: Video S5. Single cell analysis through PI staining with EFT-only

SI Line 44: Video S6. Single cell analysis through PI staining with EFT-Cu after pulses are removed

SI Line 163: 2.5 Imaging and data processing through MATLAB

SI Line 164: Traditional PI staining was still used for all the experiments and results presented, but due to the varying operation needs for the time series observation, the dye staining method was slightly different for this part of the study. To serve the purpose of the time series observation, PI stain was added before the experimentation. The EFT occurred only within the first 20 seconds of the entire experiment, but some cells might be stained due to reversible electroporation. Since there was no shift observed in the EFT-only conditions after initial pulses were removed, we are confident if there was any reversible electroporation present it could only occur in the first 1 minute of the experiment. In addition to this, the EFT conditions shown in Figure 3 where stain was added before treatment (specific to the time stamp of 2 hours) resulted in similar and consistent data to those of the same conditions in Figure 2 where dye was added 2 hours later, only at the end of treatment. This indicates further that the PI staining due to reversible electroporation is not significant and therefore not of high concern. Because of this, we consider the cells stained with PI as inactivated for the bulk of the results presented in Figure 3. Regarding the single cell study and results presented in Figure 4, the stain is also introduced before the experiment and pulse application. However, this part of the study is only focused on the individual cell, its cell membrane permeability, and initial rate of the observed fluorescence. Because of this, we are not concerned with the reversible pore closure or inactivation efficiency. We reference the concerns for reversible electroporation here to the reader for full context and understanding of the staining methods.

SI Line 202: **Video S1 | PI staining of cells treated with Cu-only.** This snapshot from Video S1 using no EFT and only Cu ion dosage (2 mg/L) shows a zoomed in portion of the channel at the center observed for ~1 min. At the top displays the relative time for each channel image. PI_2 represents the time in seconds at which each fluorescent image was taken and DIC represents the time each differential interference contrast image was taken. Specific cells that are observed to be damaged and therefore permeable to PI-staining during this minute are circled in yellow.

SI Line 215: **Video S2 | PI staining of cells treated with EFT-only.** This snapshot from Video S2 uses EFT (2 μ s pulse width, 2 ms period, 10k pulses, and 50 V) and Cu dosage (2 mg/L) to show a zoomed in portion of the channel at the center observed for ~1 min. The initial pulse application portion of both EFT-only and EFT-Cu are observed to be very similar, thus we only display one here. At the top displays the relative time for each channel image. PI_2 represents the time in seconds at which each fluorescent image was taken and DIC represents the time each differential interference contrast image was taken. Specific cells that are observed to be damaged and therefore permeable to PI-staining during the initial quick pulse application are circled in yellow.

SI Line 228: **Video S3 | PI staining of cells treated with EFT-Cu after pulses are removed.** This snapshot from Video S3 uses EFT (2 μ s pulse width, 2 ms period, 10k pulses, and 50 V) and Cu dosage (2 mg/L) to show a zoomed in portion of the channel at the center a few minutes after pulses are removed. At the top displays the relative time for each channel image. PI_2 represents the time in seconds at which each fluorescent image was taken and DIC represents the time each differential interference contrast image was taken. Specific cells that are observed to be damaged and therefore permeable to PI-staining post pulse application and by Cu ion permeation are circled in yellow.

SI Line 241: **Video S4 | Single cell analysis through PI staining with Cu-only.** This snapshot from Video S4 shows the single cell staining of Cu-only condition over ~1 min. At the top displays the relative time for each channel image. PI_2 represents the time in seconds at which each fluorescent image was taken and DIC represents the time each differential interference contrast image was taken. The specific cell that is observed to be damaged and therefore permeable to PI-staining by Cu ion permeation is circled in yellow.

SI Line 255: **Video S5 | Single cell analysis through PI staining with EFT-only.** This snapshot from Video S5 shows the single cell staining of EFT-only condition over a few seconds. At the top displays the relative time in seconds for each fluorescent image taken. The specific cell that is observed to be damaged and therefore permeable to PI-staining by rapid electroporation is circled in yellow.

SI Line 268: **Video S6 | Single cell analysis through PI staining with EFT-Cu after pulses are removed.** This snapshot from Video S6 shows the single cell staining of EFT-Cu condition after pulses are removed. At the top displays the relative time for each channel image. PI_2 represents the time in seconds at which each fluorescent image was taken and DIC represents the time each differential interference contrast image was taken. The specific cell that is observed to be damaged and therefore permeable to PI-staining by Cu ion permeation; post pulse application is circled in yellow.

REFERENCES

- 1 Novickij, V. *et al.* Low concentrations of acetic and formic acids enhance the inactivation of *Staphylococcus aureus* and *Pseudomonas aeruginosa* with pulsed electric fields. *BMC Microbiology* **19**, 73 (2019). <https://doi.org/10.1186/s12866-019-1447-1>
- 2 Davey, H. M. & Hexley, P. Red but not dead? Membranes of stressed *Saccharomyces cerevisiae* are permeable to propidium iodide. *Environ Microbiol* **13**, 163-171 (2011). <https://doi.org/10.1111/j.1462-2920.2010.02317.x>
- 3 Wang, T. & Xie, X. Nanosecond bacteria inactivation realized by locally enhanced electric field treatment. *Nature Water* **1**, 104-112 (2023). <https://doi.org/10.1038/s44221-022-00003-2>
- 4 Wang, T., Brown, D. K. & Xie, X. Operando Investigation of Locally Enhanced Electric Field Treatment (LEEFT) Harnessing Lightning-Rod Effect for Rapid Bacteria Inactivation. *Nano Letters* **22**, 860-867 (2022). <https://doi.org/10.1021/acs.nanolett.1c02240>
- 5 Kotnik, T., Rems, L., Tarek, M. & Miklavčič, D. Membrane electroporation and electropermeabilization: mechanisms and models. *Annual review of biophysics* **48**, 63-91 (2019).
- 6 Stiefel, P., Schmidt-Emrich, S., Maniura-Weber, K. & Ren, Q. Critical aspects of using bacterial cell viability assays with the fluorophores SYTO9 and propidium iodide. *BMC Microbiology* **15**, 36 (2015). <https://doi.org/10.1186/s12866-015-0376-x>
- 7 Napotnik, T. B., Polajžer, T. & Miklavčič, D. Cell death due to electroporation—a review. *Bioelectrochemistry* **141**, 107871 (2021).
- 8 Wilks, S., Michels, H. & Keevil, C. The survival of *Escherichia coli* O157 on a range of metal surfaces. *International journal of food microbiology* **105**, 445-454 (2005).
- 9 Quaranta, D. *et al.* Mechanisms of contact-mediated killing of yeast cells on dry metallic copper surfaces. *Applied and environmental microbiology* **77**, 416-426 (2011).
- 10 Santo, C. E. *et al.* Bacterial killing by dry metallic copper surfaces. *Applied and environmental microbiology* **77**, 794-802 (2011).
- 11 Hong, R., Kang, T. Y., Michels, C. A. & Gadura, N. Membrane lipid peroxidation in copper alloy-mediated contact killing of *Escherichia coli*. *Applied and environmental microbiology* **78**, 1776-1784 (2012).
- 12 Warnes, S., Caves, V. & Keevil, C. Mechanism of copper surface toxicity in *Escherichia coli* O157: H7 and *Salmonella* involves immediate membrane depolarization followed by slower rate of DNA destruction which differs from that observed for Gram-positive bacteria. *Environmental microbiology* **14**, 1730-1743 (2012).
- 13 Kotnik, T. *et al.* Electroporation-based applications in biotechnology. *Trends in biotechnology* **33**, 480-488 (2015).
- 14 Li, J. & Lin, H. Numerical simulation of molecular uptake via electroporation. *Bioelectrochemistry* **82**, 10-21 (2011). [https://doi.org:https://doi.org/10.1016/j.bioelechem.2011.04.006](https://doi.org/https://doi.org/10.1016/j.bioelechem.2011.04.006)
- 15 Weaver, J. C. & Chizmadzhev, Y. A. Theory of electroporation: a review. *Bioelectrochemistry and bioenergetics* **41**, 135-160 (1996).
- 16 Huo, Z.-Y. *et al.* Nanowire-modified three-dimensional electrode enabling low-voltage electroporation for water disinfection. *Environmental science & technology* **50**, 7641-7649 (2016).
- 17 Catterall, W. A. Structure and regulation of voltage-gated Ca²⁺ channels. *Annual review of cell and developmental biology* **16**, 521-555 (2000).
- 18 Garner, A. L. Pulsed electric field inactivation of microorganisms: from fundamental biophysics to synergistic treatments. *Applied Microbiology and Biotechnology* **103**, 7917-7929 (2019). <https://doi.org/10.1007/s00253-019-10067-y>
- 19 Arndt-Jovin, D. J. & Jovin, T. M. Fluorescence labeling and microscopy of DNA. *Methods in cell biology* **30**, 417-448 (1989).

REVIEWERS' COMMENTS

Reviewer #1 (Remarks to the Author):

The authors have addressed my comments.

Reviewer #2 (Remarks to the Author):

As the authors point out, their setup makes determining microorganism viability by standard techniques challenging, leading them to the application of PI. As I mentioned in my previous review, I have seen similar behavior with other stains where introducing the stain a sufficiently long time after treatment would eliminate the effects of the long-lived pores in assessing the implication of dye uptake for viability.

The authors have described this behavior clearly in the response and cleared it up in the manuscript. The main approach of this manuscript is interesting. While there may be an alternative way to tease out the viability specifically, they have clarified their thought process on how they did this sufficiently for a reader to make their own conclusion. In my mind, this makes the manuscript worth publication at this point.